

# Detecting local communities in complex network *via* the optimization of interaction relationship between node and community

Shenglong Wang[1], Jing Yang[1], Xiaoyu Ding[2] and Meng Zhao[1]

[1] College of Computer Science and Technology, Harbin Engineering University, Harbin, China
[2] Chongqing University of Posts and Telecommunications, Chongqing, China

## ABSTRACT

The goal of local community detection algorithms is to explore the optimal community with a reference to a given node. Such algorithms typically include two primary processes: seed selection and community expansion. This study develops and tests a novel local community detection algorithm called *OIRLCD* that is based on the optimization of interaction relationships between nodes and the community. First, we introduce an improved seed selection method to solve the seed deviation problem. Second, this study uses a series of similarity indices to measure the interaction relationship between nodes and community. Third, this study uses a series of algorithms based on different similarity indices, and designs experiments to reveal the role of the similarity index in algorithms based on relationship optimization. The proposed algorithm was compared with five existing local community algorithms in both real-world networks and artificial networks. Experimental results show that the optimization of interaction relationship algorithms based on node similarity can detect communities accurately and efficiently. In addition, a good similarity index can highlight the advantages of the proposed algorithm based on interaction optimization.

## INTRODUCTION

Currently, the development of information technology has to the emergence of various complex networks, enriching application scenarios such as activist groups, schoolmate discovery, protein function identification and e-commence recommendation (*Fang et al., 2020*). Identifying the structure of communities is one of the most important fields in the research of complex networks and has attracted the attention of many researchers to participate (*Mittal & Bhatia, 2020*). In today's world, there exists various community structures, which consist of different types of entities, called nodes, and the connections between these entities are known as links (*Pizzuti, 2018*). Nodes within the same community are closely connected, while nodes between different communities are sparsely connected (*Garza & Schaeffer, 2019*).

In recent years, researchers have paid more attention to the study of community detection. The detection of community structure can help discover various groups in

Corresponding authors
Shenglong Wang,
wangsl@hrbeu.edu.cn
Jing Yang, yangjing@hrbeu.edu.cn

society, which in turn help people solve real-world problems. Local community detection aims to detect communities using only local topological information of nodes. This approach has a lower time complexity and is more convenient for accessing to information in complex networks than the approaches using global topological information.

Local community detection algorithms typically use a node as a seed and expand from this seed to identify a community based on optimizing a quality function (*Kanawati, 2015*; *Zhang et al., 2015*; *Peng & Jing, 2016*; *Liakos, Ntoulas & Delis, 2016*; *Zhu, Chen & Zeng, 2020*). The seed selection process and community expansion processes play critical roles in local community detection algorithms, and they have a significantly impact on the quality of resulting communities. Unfortunately, there are still problems that hinder the development of research in local community detection in terms of these two areas. First, the quality of the resulting communities detected by algorithms heavily depends on the seed node selected at the beginning, which is known as the seed dependence problem (*Ding, Zhang & Yang, 2018*). Second, some algorithms (*Ding, Zhang & Yang, 2018*; *Lee et al., 2010*; *Li, Wang & Cui, 2014*; *Cheng et al., 2020*; *Luo et al., 2017*; *Ni et al., 2020*; *Malliaros & Vazirgiannis, 2013*) search for alternative seed nodes that are more suitable for community expansion than the given node. However, the alternative seed node and the given node are not always in the same community, resulting in what is called the seed deviation problem. Third, the structural characteristics of resulting communities detected by algorithms are limited by the quality function, which is known as the quality function limitation problem (*Ding, Zhang & Yang, 2020*).

To address the first and second problems, this study presents an improved seed selection method called *SSCS* based on node centrality and node similarity. This novel seed selection method identifies the most similar neighbor node of a given node, which has higher node centrality than the given node, and takes this node as the alternative seed of the given node. This process is repeated iteratively until there are no neighbors that meet the above two conditions, and the final result is taken as the seed. To address the third problem, this study uses a novel local community detection algorithm called *OIRLCD*. It optimizes the interaction relationships between nodes and communities, also known as the interaction relationship, by deprecating the quality function. We introduced a series of similarity indices to measure the interaction relationship between nodes and communities and expanded communities by adding the node with the most important interaction relationship to the community.

The primary contributions of this article can be summarized as follows.

- To address the seed dependence and deviation problems, this study develops an improved seed selection method based on node centrality and node similarity. The method identifies the core node of the community that the given node locates as the alternative seed. First, this method first compares the similarity between the target node and its neighbors, and then compares the node centrality between the target node and its neighbors. This process ensures that the alternative seed and the target node are in the same community as much as possible.

- To measure the interaction relationship, this study uses a series of node similarity indices based on the local topological information of nodes. We also investigated the role of the similarity index in local community detection algorithms based on interaction relationships. To this end, we designed a series of similarity indices with various amounts of local topological information of nodes. We compared these indices with the other three basic similarity indices and the three latest similarity indices respectively under the same framework.

- To avoid the quality function limitation problem, this study proposes a novel local community detection algorithm based on the optimization of interaction relationships, which leverages the seed selection method and community expansion method earlier.

- We compared the proposed algorithm with different similarity indices on three groups of artificial networks and six real-world networks. Experimental results show that the proposed seed selection method can improve the accuracy of the algorithm; the proposed algorithm outperforms six existing community detection algorithms; and a good similarity index can highlight the advantages of algorithms based on interaction optimization.

The remainder of this article is outlined as follows. Related research of seed selection methods, community expansion and similarity indices are described in the "Related Works". "Motivations and Basic Definitions" presents the definitions related to this study and the detailed procedures of the proposed algorithm. "Experiments and Analysis" expounds on the experimental process and results in detail, and the results are analyzed. Finally, in "Conclusion", we concluded this study and outlook for the future research.

## RELATED WORKS

The seed selection process and community expansion process are two critical steps in the local community detection algorithms. A good seed selection method can lead to high-quality seeds, which improves algorithms accuracy and efficiency. A good community expansion method can efficiently identify node membership, generating the resulting community quickly and correctly. A good similarity index can accurately measure the relationships between two nodes, or between nodes and communities within low time complexity. This section introduces the latest methods related to the seed selection method and the community expansion method and similarity indices and shows their characteristics.

### Seed selection

The goal of the seed selection method is to identify the core node of the community where the target node is located, which can improve the quality of the initial community (*Wang et al., 2016*). To obtain high-quality seeds as the initial community for expansion, a variety of seed selection methods, had been proposed by scholars. *Lancichinetti, Fortunato & Kertész (2009)* used a random selection method which is the simplest and most time-saving method to select nodes as seeds. However, the random selection method will make the algorithm unstable, which results in uncontrollable results. Similarly, *Baumes et al. (2005)*

also used a random selection method, but instead of selecting random nodes, they replaced random edges as seeds. However, searching for random edges as seeds will generate many duplicate communities, which will lead to an increase in the algortithm's time complexity of the algorithm and thus require a lot of computation time. *Lee et al. (2010)* explored k-clique, which is a complete subgraph with k vertices, of the target node as seeds. Based on the seed selection method, *Lee et al. (2010)* proposed a *Greed Clique Expansion (GCE)* algorithm. Furthermore, *Li, Wang & Cui (2014)* took maximum cliques as the seed, searched using depth and breadth search methods, and merged different communities into a larger sub-graph according to the given rules. To eliminate the influence of the seed quality on the local community detection algorithm, *Ding, Zhang & Yang (2018)* proposed a core member searching method that iteratively replaces the initial node with the candidate seed that has greater local influence and is most similar to the given node. *Cheng et al. (2020)* ranked nodes of networks according to the Technique for Order of Preference by Similarity to Ideal Solution (*TOPSIS*), and the node with the highest score was used as the seed. *Ni et al. (2020)* took with the NGC node (*Luo et al., 2017*), the nearest node with the greater centrality, and selected nodes with greater fuzzy relationships among their *NGC* nodes are considered to be the seeds.

## Community expansion

The goal of the community expansion is to expand the initial community into the resulting community through an expansion mechanism. The commonly used expansion mechanisms are the quality function (*Kanawati, 2015*; *Zhang et al., 2015*; *Peng & Jing, 2016*; *Liakos, Ntoulas & Delis, 2016*; *Zhu, Chen & Zeng, 2020*) and influence spreading (*Kloster & Gleich, 2014*; *Hu, Yang & Wong, 2016*; *He et al., 2015*; *Yao et al., 2016*; *You, Ma & Liu, 2020*). The quality function is a measure of the quality of community division results derived from the definition of community structure. *Newman & Girvan (2004)* proposed modularity as the quality of the community for measuring the community quality. According to the definition of modularity, high-quality communities should have a tight internal structure and loose external links between communities. *Guo et al. (2022)* proposed an improved algorithm that takes the which take local modularity density as the quality function.

The influence spreading method expands the community by calculating the influence of nodes and spreading these influences throughout the network. *Raghavan, Albert & Kumara (2007)* proposed the Label Propagation algorithm (*LPA*) based on an epidemic spreading model. *LPA* assigns each node of the network a unique label and spreads these labels over the entire network. *Xu, Guo & Yang (2020)* proposed a novel similarity measure based on a two-level neighborhood (*TNS*). Using *TNS* as a basis, they also proposed an improved *LPA* algorithm.

## Similarity index

Nodes within the same community exhibit high similarity, whereas those between communities are not typically similar (*Malliaros & Vazirgiannis, 2013*). Therefore, similarity index can also measure the memberships between nodes and communities.

**Table 1 Common used similarity indices.**

| Similarity index name | Definition | Formula | References |
|---|---|---|---|
| Jaccard index | The ratio of the intersection of two nodes' neighbors to the union of two node's neighbors. | $\frac{\|N(v_i) \cap N(v_j)\|}{\|N(v_i) \cup N(v_j)\|}$ | Jaccard (1901) |
| Salton index | The ratio of the intersection of two nodes' neighbors to the radical sign of the product of the number of two nodes' neighbors. | $\frac{\|N(v_i) \cap N(v_j)\|}{\sqrt{\|N(v_i)\|\|N(v_j)\|}}$ | Salton & McGill (1986) |
| Resource allocation (RA) index | The sum of the reciprocal of degrees of all nodes within the intersection of two node neighbors. | $\sum\limits_{v_n \in N(v_i) \cap N(v_j)} \frac{1}{d_{v_n}}$ | Zhou, Lü & Zhang (2009) |
| Common neighbors (CN) index | The size of intersection of two node neighbors. | $\|N(v_i) \cap N(v_j)\|$ | Granovetter (1973) |

*Ding, Zhang & Yang (2020)* proposed the local expansion and boundary rechecking (*LEBR*) algorithm, which optimized the membership between nodes and communities to expand communities, rather than optimizing the quality function. *Ding, Zhang & Yang (2020)* demonstrated that *LEBR* is highly effective at detect communities with diverse structures; thus, the limitation problem caused by the quality function is avoided. Table 1 displays commonly used node similarity indices.

In recent years, scholars have proposed various node similarity indices, leading to progress in node similarity calculation accuracy. The similarity indices related to this article are as follows.

*Zhang, Ding & Yang (2019)* reported that the similarity between two adjacent nodes increases as their *k*-core value grows larger. To distinguish between external and internal nodes of the community, they introduced the concept of local *k*-core value in their algorithm. Furthermore, the contribution of two adjacent nodes to their similarity should be different. The core similarity (*CS*) between two nodes is defined as follows.

$$\mathbf{S}(v_i, v_j) = \frac{\mathbf{K}_{N(v_i) \cap N(v_j)}(v_i)}{\mathbf{K}_V(v_i)} \sqrt{\mathbf{K}_V(v_i) * \mathbf{K}_V(v_j)} \tag{1}$$

where $\mathrm{K}_{N(v_i) \cap N(v_j)}(v_i)$ is the local *k*-core value of node $v_i$ in the interaction of neighborhood of $v_i$ and $v_j$, $\mathrm{K}_V(v_i)$ is the *k*-core value of node $v_i$ in the whole network.

Inspired by the *RA* index and local path (*LP*) similarity index (*Zhou, Lü & Zhang, 2009*), *Xu, Guo & Yang (2020)* proposed a novel similarity index based on a two-level neighborhood of nodes. *RA* makes full use of the topological information of nodes to improve the accuracy of similarity between nodes. *LP* similarity index and the two-level neighborhood similarity (*TNS*) index are defined as follows.

$$\mathbf{S} = \mathbf{A}^2 + \alpha \mathbf{A}^3 \tag{2}$$

where S denotes the similarity matrix, A denotes the node adjacent matrix and $\alpha$ denotes the free parameter.

$$S(v_i, v_j) = \sum_{v_l \in N(v_i) \cap N(v_j)} \frac{1}{d_{v_l}} + \sum_{v_m \in N(v_i),\, v_n \in N(v_j)} \frac{A_{mn}}{\sqrt{d_{v_m} d_{v_n}}} \qquad (3)$$

*Liu et al. (2022)* introduced a node similarity index named *CN*, which combines the common neighbors and degree of node. *CN* is defined as follows.

$$S(v_i, v_j) = |N(v_i) \cap N(v_j)| + \frac{1}{d_{v_i} d_{v_j}} \qquad (4)$$

# MOTIVATIONS AND BASIC DEFINITIONS

## Motivation

As described in "Related Works", researches in the field of seed selection methods, community expansion methods and similarity indices have made a lot of progress. However, there are still problems with the implementation of local community detection algorithms, which prevent accurate results from being obtained.

In the realm of community detection algorithms, one of the most significant challenges is the seed dependence problem. Essentially, the quality of the given node determines the accuracy of the resulting community partition. To address this issue, *Ding, Zhang & Yang (2018)* proposed a seed selection method called *SSSC* that effectively solves the seed dependence problem. This problem arises when the accuracy of the community detection algorithm depending heavily on the quality of the given seed. Specifically, the method involves comparing the centrality between the alternative seed and the given node, and then comparing the similarity between these two nodes with the maximum similarity obtained before. However, this method is correct only when the alternative seed and the given node are in the same community. In cases where the alternative seed and the given node are not in the same community, but the alternative seed has the greatest centrality and greater similarity with the given node, *SSSC* will still consider this node as the alternative seed of the given node. This leads to incorrect results, which we refer to as the seed deviation problem. As such, further research is required to address this issue and improve the accuracy of community detection algorithms.

Secondly, a local community detection algorithm typically optimize only one type of quality function during the process of community expansion. While this approach my yield satisfactory results for certain types of networks, it can lead to less efficient performance when dealing with other types of networks. In particular, a quality function that describes a community with only one structural feature may not be sufficient for more complex networks (*Ding, Zhang & Yang, 2020*). As a result, community detection algorithms may face the quality function limitation problem. As such, further research is required to address this issue and improve the accuracy of community detection.

To solve the problems of seed dependence and seed deviation, we propose an improve seed selection method that first considers similarity first. This ensures that the alternative seed and the given node are in the same community. We then calculate the node centrality. We consider the problem with a simple example in Fig. 1. As shown in Fig. 1, the similarity between $v_3$ and $v_1$ is lower than that between $v_2$ and $v_1$, but $v_3$ has a greater node centrality

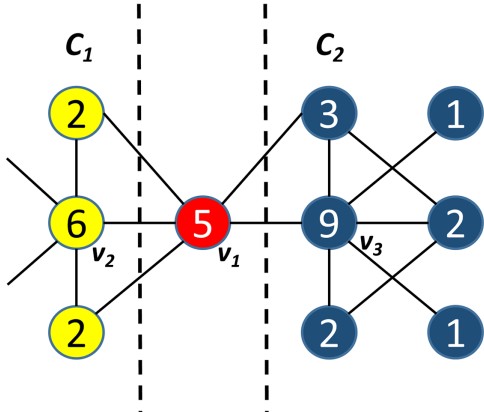

**Figure 1** **A sample of seed selection method.** $v_1$ is the given node with node centrality 5, $v_2$ is the given node with node centrality 6, $v_3$ is the given node with node centrality 9.

(9) than $v_2$, which has a centrality of (6). In this condition, *SSSC* considers $v_3$ as the alternative seed of $v_1$ under this condition. However, since $v_2$ is more similar to $v_1$ than $v_3$ is. Additionally, $v_3$, and $v_1$ are in the same community. Therefore, $v_2$ is actually the alternative seed of $v_1$.

To mitigate the quality function limitations problem, a novel local community detection algorithm based on the optimization of the interaction relationships is proposed, which deprecates the quality function. The proposed motivation is to develop with precision similarity indices that can accurately calculate the interaction relationship. To achieve higher precision in measuring node similarity than existing measures, this study gradually obtains more neighbourhood information gradually.

## Problem definition

This study focuses on a graph called $G = (V, E)$. The node set composed of all nodes in the graph is represented by $V$. The link set composed of all links between these nodes is represented by $E$, and $A$ is a two-dimensional array called adjacent matrix that records whether two nodes are connected. $A_{ij} = 1$ denotes that there is a link between node $i$ and node $j$ that is connected; otherwise, $A_{ij} = 0$.

The given node denotes an initial node given in local community detection algorithms. A community $C$ denotes a collection of nodes and their connected links, where C = {$v_1$, $v_2$, …, $v_j$} (C ∈ **C**, $v_i$ ∈ V). The initial community denotes the community composed of seed and its part of neighbors. The expending community denotes the community in expanding. The result community denotes the community detected by algorithms. This study aims to detect a community **C** where the given node really locates.

## Basic definitions

**Definition 1** (Node neighbors). The node neighbors of node v are defined as follows:

$$N(v) = \{u|u \in V, A_{uv} = 1\}, \ v \in V \tag{5}$$

where A is the adjacent matrix of graph G, and if $A_{uv} = 1$, it means that there is a link

between node $v$ and node $u$. The definition of node neighbors is a set of nodes with links connected to the node.

**Definition 2** (Node influential scope). The node influential scope of node v is defined as follows:

$$NI(v) = \{u|u \in N(v), A_{uv} = 1\}, \ v \in V \tag{6}$$

where $N(v)$ denotes node neighbors defined in Definition 1. The definition of node influential scope is a set of nodes consists of node neighbors and node itself.

**Definition 3** (Community neighbors). The community neighbors of community C is defined as follows:

$$N(C) = \{u|u \notin C, N(v), \exists v \in C, (u,v) \in E\}, \ v \in V, \{u|u \in |C, \exists v \in C, (u,v) \in E\} \tag{7}$$

where E denotes the set of links of network G. The defininlition of community neighbors is a set of external nodes that have links connected to the members of the community.

**Definition 4** (Node degree). The node degree of node v is defined as follows:

$$d(v) = |N(v)|, \ v \in V \tag{8}$$

The definition of node degree is the number of node links.

**Definition 5** (Local centrality). The local centrality of node v is defined as follows:

$$LC(v) = \{v_i, v_j|v_i, v_j \in NI(v), \ A_{ij} = 1\}, \ v \in V \tag{9}$$

We measure node centrality by examining links within the node's influential scope defined in Definition 2. The more links there are within the scope, the greater the node's centrality.

**Definition 6** (Node similarity 1). The first similarity index proposed in this article between node $v_i$ and $v_j$ is defined as follows:

$$S1(v_m, v_n) = \{v_i, v_j|v_i, v_j \in N(v_m) \cap N(v_n), \ A_{ij} = 1\}, \ v \in V \tag{10}$$

We measure the similarity between the neighborhood of $v_m$ and $v_n$ by analyzing the links between them. The more links there are within the two nodes' influential scope, the greater their similarity. We can describe this similarity index with the simple example shown in Fig. 2, where $S1(v_1, v_2) = 14$.

**Definition 7** (Node similarity 2). The second similarity index proposed in this article between node $v_i$ and $v_j$ is defined as follows:

$$S2(v_m, v_n) = \sum\nolimits_{v_i, v_j \in N(v_m) \cap N(v_n), \ A_{ij}=1} |d(v_i) + d(v_j)|, \ v \in V \tag{11}$$

The contribution of each link within the node influential scope to the similarity of two nodes is likely not the same. Therefore, we assign weights to the links based on the degree of nodes on both sides of the link. The similarity between nodes is then calculated as the degree sum of the nodes at both ends of the link within the common influence scope.

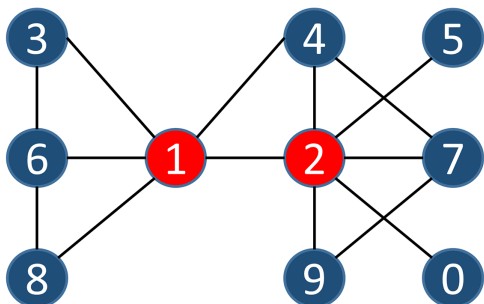

**Figure 2** A sample of a simple network.

**Definition 8** (Node similarity 3). The third similarity index proposed in this article between node $v_i$ and $v_j$ is defined as follows:

$$S3(v_m, v_n) = \sum\nolimits_{v_l \in N(v_m) \cap N(v_n)} |NI(v_l)| * S2(v_m, v_n), \; v \in V \tag{12}$$

The contribution of each node within the node influential scope compared to the similarity of two nodes is likely not the same. Thus, Definition 7 describes the similarity index within the influence scope of these two adjacent nodes. Based on Definition 7, we multiply the number of nodes within the influence scope of each node in the common influential scope of two adjacent nodes by the similarity index, and sum all that of nodes in the scope. We can show this similarity index with the simple example in Fig. 2, where $S1(v_1, v_2) = (3 + 5 + 3 + 7 + 5 + 9 + 3 + 1 + 5 + 1) * 102 = 4{,}284$.

**Definition 9** (Node community similarity). The node community similarity between node v and community C is defined as follows:

$$NCS(v, C) = \sum\nolimits_{u \in (N(v) \cap C)} NS(u, v), \; u, v \in V \tag{13}$$

where $NS(u, v)$ represents a method of node similarity calculation.

We calculate the similarity between node $v$ and community $C$ by sum the similarity between node $v$ and each node in community which has a link with node $v$.

## Proposed algorithm

Algorithm 1 shows the pseudocode of *OIRLCD*. To facilitate readers' understanding of the proposed algorithm, we provide flowcharts of the seed selection process and community expansion process in Figs. 3 and 4, respectively. This section provides a detailed description of the proposed algorithm.

*Initialization* (Lines 1–4). Line 2 initializes the empty community C, which will store the final result. Based on Definition 4, Line 3 calculates the node degree of each node in node set V. Based on Definition 5, Line 4 calculates the local centrality of each node in node set V.

*Seed selection* (Lines 4–21). The seed selection process searches for the core node of the community where the given node is located as the alternative seed. To find the alternative seed for a given node, two requirements must be met. First, the alternative seed must have the maximum similarity to the given node to ensure that they are in the same community.

**Algorithm 1** **The local community detection algorithm based on optimization of interaction relationship (OIRLCD)**

**Input**: Graph $G = <V, E>$, link set E, node set V, seed node $v_{seed}$, Node influence measure Inf.

**Output**: Community $C$.

**Process**:

1: **Initialization:**

2: Initialize a community $C$, $C = \phi$;

3: Calculate the degree $d(v_i)$ of each node based on Definition 4, $v_i \in V$;

4: Calculate the local centrality $LC(d_{v_i})$ of each node based on Definition 5, $v_i \in V$;

5: **Seed selection process:**

6: Set $v_{temp} = v_{seed}$;

7: Set $max\_similarity = 0$;

8: **do**

9:  Initialize $v_{seed} = v_{temp}$;

10:  Calculate neighboring nodes $N(v_{temp})$ of $v_{temp}$ based on **Definition** 1;

11:  **for all** $v_i \in N(v_{temp})$ **do**

12:   Calculate the node similarity $S(v_i, v_{temp})$ between $v_i$ and $v_{temp}$ based on Definition 8;

13:   **if** $S(v_i, v_{temp}) > max\_similarity$ **then**

14:    $max\_similarity = S(v_i, v_{temp})$;

15:    **if** $LC(v_i) > LC(v_{temp})$ **then**

16:     $v_{temp} = v_i$;

17:    **end if**

18:   **end if**

19:  **end for**

20: **while** $v_{seed} \neq v_{temp}$

21: return $v_{seed}$;

22: **Community expansion process:**

23: initialize $C = NI(v_{seed})$;

24: cleanup $C$.

25: initialize $C_{temp} = C$;

26: Set $suspicious\_list = \phi$;

27: **do**

28: $C = C_{temp}$;

29: Get the community neighbors $N(C_{temp})$ based on Definition 3;

30: $suspicious\_list = N(C)$;

31: **while** $suspicious\_list \neq \phi$ **do**

32:  Pull $v_i$ from $suspicious\_list$;

33:  Calculate the node community similarity $NCS(v_i, C)$ between $v_i$ and $C$ based on Definition 9;

**Algorithm 1** (continued)

34:  Calculate the node community similarity $NCS(v_i, \bar{C})$ between $v_i$ and $\bar{C} = N(v_i) - C$ based on Definition 9;

35:  **if** $NCS(v_i, C) > NCS(v_i, \bar{C})$ **then**

36:   add $v_i$ to $C$;

37:  **end if**

38:  **end while**

39: **while** $C \neq C_{\text{temp}}$

40: **return** $C$

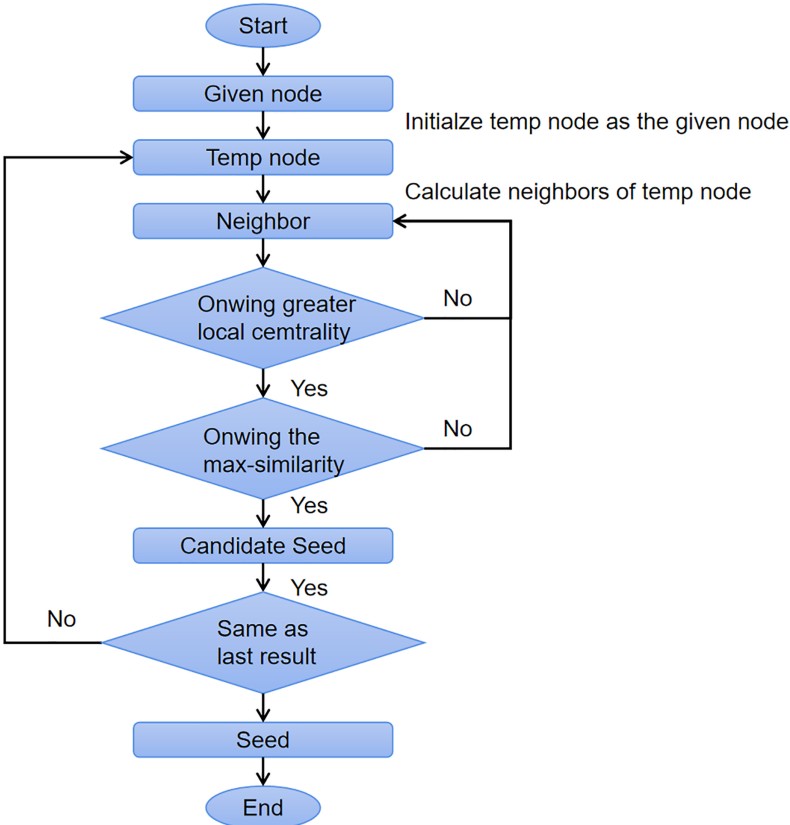

**Figure 3 The flow chart of seed selection process.**   

Second, the alternative seed must have the greater local centrality than the given node to ensure that the alternative seed is closer to the center of the community than the given node. As shown in Algorithm 1, Line 7 sets *max_similarity* as zero to store the greatest similarity value. Line 10 obtains all neighboring nodes $N(v_{temp})$ of $v_{temp}$ based on Definition 1. Line 12 calculates node similarity based on Definition 8 to ensure that all the comparison algorithms with different similarity indices should have the same seed node for community expansion. For each node in $N(v_{temp})$, the process will replace the previous

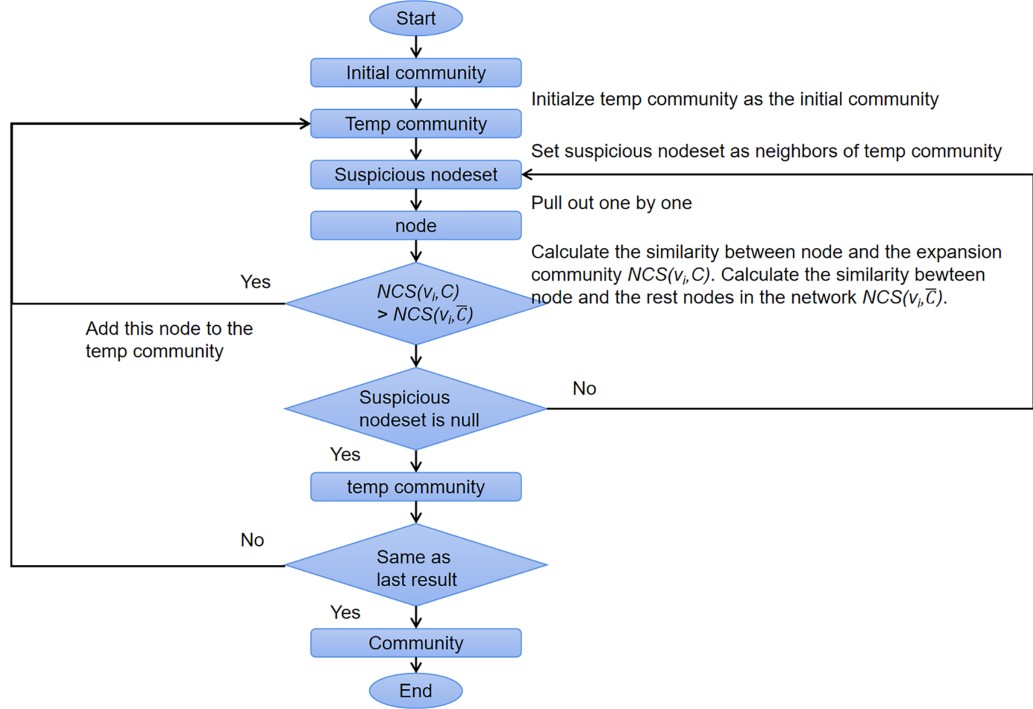

**Figure 4 The flow chart of community expansion process.**

alternative seed (Line 16) if it satisfies the two conditions mentioned above. Executing the program until all nodes in $N(v_{temp})$ are calculated (Lines 11–19). Lines 8–20 search for the alternative seed until no neighboring nodes of the current alternative seed meet the conditions. Line 20 sets the current alternative seed as the seed and sends this seed to the community expansion process.

*Community expansion* (Lines 21–39). For community expansion, the proposed algorithm gradually adds the community neighbors that meet specific conditions. An eligible community neighbor is one whose similarity to the community is greater than its similarity to the rest of the nodes in the network. As shown in Algorithm 1, Line 23 initializes the initial community $C$ as the influential scope of the seed. Line 24 excluded nodes which is not meeting the conditions above. Line 29 obtains all community neighbors $N(C_{temp})$ of $C_{temp}$ based on Definition 3. Line 33 calculates the node community similarity $NCS(v_i, C)$ based on Definition 9 between node $v_i$ and the community $C$. The remaining nodes of the network G are regarded as community $\bar{C}$. Line 34 calculates the node community similarity $NCS(v_i, \bar{C})$ between node $v_i$ and the community $\bar{C}$. When $NCS(v_i, C) > NCS(v_i, \bar{C})$, $v_i$ should be added to the community $C$. Executing the program until all nodes in $N(C_{temp})$ are calculated (Lines 31–38). Lines 27–39 execute community expansion until no community neighboring nodes of the current community $C$ meet the conditions. Line 20 returns the current community $C$.

## Time complexity analysis

This section analyzes the time complexity of *OIRLCD*. All the comparison algorithms are executed on the network $G = (V, E)$ where the average degree $\bar{d}$.

First, the initialization process includes Definition 4 and Definition 5, with the time complexity of $O(\bar{d})$ and $O(\bar{d}^2)$. So the first process needs $O(\bar{d}^2)$. Secondly, the seed selection process use the Definition 8 to calculate the node similarity, and its time complexity is $O(\bar{d}^2)$. So the seed selection process needs $O(\bar{d}^3)$. Thirdly, the community expansion process first calculates the community neighbors based on Definition 3 with time complexity $O(\bar{d}^2)$. The mean distance from the community edge to its core node is set as $\bar{r}$. The proposed similarity index defined as Definition 6, Definition 7 and Definition 8 with time complexity $O(\bar{d}^2)$, $O(\bar{d}^2)$, $O(\bar{d}^2)$, respectively. Finally, the overall time complexity of our three proposed algorithm need $O(\bar{r}\bar{d}^2)$.

Calculating Jaccard similarity index needs $O(1)$, calculating Salton similarity index needs $O(1)$, calculating *RA* similarity index needs $O(\bar{d})$, calculating the *CS* similarity index needs $\mathrm{O}(n\bar{d}^3)$, calculating the *TNS* similarity index needs $\mathrm{O}(n\bar{d}^2)$ and calculating the *CN* similarity index needs $\mathrm{O}(\bar{d}^2)$. Finally, the overall time complexity of the proposed algorithms and the comparison algorithms are listed in Table 2. The symbols in the table that $C$ denotes the size detected community; $\overline{|C|}$ denotes the size of detected community; $|S|$ denotes the size of the shell sub-network of $C$; $|N|$ denotes the size of the neighbor sub-network of $C$.

## EXPERIMENTS AND ANALYSIS

The experimental environment of this study is as follows: the proposed algorithm and the comparison algorithms are programmed in *JAVA*; all the programs involved in this study are running in a computer with AMD Ryzen 5 5600H with Radeon Graphics 3.30 GHz and 16 GB RAM. The experiments are implemented in the proposed algorithm and seven comparison algorithms on six real-world networks and three groups of different parameters artificial networks, and the experimental results using four commonly used local community indicators. Table 3 displays related symbols and their explanations.

## Evaluation criteria

Normalized mutual information (*Danon et al., 2005*) (NMI) and F-score (*Li, Wang & Wu, 2015*) is two widely used methods for evluating community quality. This study verified the resulting communities of *OIRLCD* and comparison algorithms on these two indicators.

### Normalized mutual information

*Danon et al. (2005)* used information entropy to measure the quality of a cluster. This information entropy describes the uncertainty of possible events of an information source They called this method the normal mutual information (*NMI*) measure (*Danon et al., 2005*). In the definition of *NMI*, matrix $N$ with rows are members from real-world communities and columns are members from the detected communities. Element $\mathbf{N}_{ij}$ in matrix $\mathbf{N}$ represent the numbers of nodes that exist in both community $i$ and community $j$ (*Danon et al., 2005*). The formula for *NMI* is as follows:

**Table 2 Time complexity list of *OIRLCD* and comparison algorithms.**

| Algorithms | Time complexity | References |
|---|---|---|
| OIRLCDF | $O(\overline{rs}\overline{d}^2)$ | [-] |
| OIRLCDS | $O(\overline{rs}\overline{d}^3)$ | [-] |
| OIRLCDT | $O(\overline{rs}\overline{d}^3)$ | [-] |
| Jaccard | $O(\overline{d})$ | *Jaccard (1901)* |
| Salton | $O(\overline{d})$ | *Salton & McGill (1986)* |
| RA | $O(\overline{d})$ | *Zhou, Lü & Zhang (2009)* |
| CS | $O(\overline{|C|}(\overline{d}log\overline{|C|}))$ | *Zhang, Ding & Yang (2019)* |
| TNS | $O(\overline{d}\overline{|C|}log\overline{|C|})$ | *Xu, Guo & Yang (2020)* |
| CN | $O(\overline{d}^2)$ | *Liu et al. (2022)* |
| LWP | $O(\overline{d}\overline{|C|}^2)$ | *Clauset (2005)* |
| Chen | $O(\overline{d}\overline{|C|}^2|N|)$ | *Chen, Zaï & Goebel (2009)* |
| LS | $O(max\{\overline{d}|N||S|,\ \overline{d}|N|log|N|\})$ | *Wu et al. (2012)* |
| LCD | $O(max\{|S|3^{\overline{d}/3}, |S|\overline{|C|}^2\})$ | *Fanrong et al. (2014)* |
| RTLCD | $O(rmax\{\overline{d}\overline{|C|}log\overline{|C|},\ \overline{|C|}(\overline{d}log\overline{|C|}) + \overline{d}^4\})$ | *Ding, Zhang & Yang (2018)* |

**Table 3 List of symbols and descriptions.**

| Symbols | Descriptions (for network *G*) |
|---|---|
| n | The number of nodes |
| m | The number of links |
| $\overline{d}$ | The mean degree |
| $d_{max}$ | The maximum degree of node |
| $\overline{|C|}_{min}$ | The minimum size of the community |
| $\overline{|C|}_{max}$ | The maximum size of the community |
| $\overline{|C|}$ | The average size of the community |
| μ | The mixing parameter |
| $O_n$ | The number of overlapping nodes |
| $O_m$ | The average number of node overlaps |
| $n_C$ | The number of communities |

$$NMI(C_A, C_B) = \frac{-2\sum_{i=1}^{|C_A|}\sum_{j=1}^{|C_B|}N_{ij}log(N_{ij}N/N_{i.}N_{.j})}{\sum_{i=1}^{|C_A|}N_{i.}log(N_{i.}/N)\sum_{j=1}^{|C_B|}N_{.j}log(N_{.j}/N)} \tag{14}$$

where $|C_A|$ denotes communities in real-world, $C_B$ denotes communities detected by the algorithms. $Ni.$ and $N.j$ denote the sums of the elements in row *i* and column *j*, respectively.

*NMI* can measure the similarity between clustering results and the real-world data. The greater the similarity between the communities detected by algorithms and the real-world communities, the higher the *NMI*. The maximum value of *NMI* is one when the results and real-world are identical.

### F-score

F-score (*Li, Wang & Wu, 2015*) is common used in the evaluation method of classification model. F-score is defined as follows:

$$F = 2 \times \frac{Precision \times Recall}{Precision + Recall} \tag{15}$$

$$Recall = \frac{C_R \cap C_D}{C_G} \tag{16}$$

$$Precision = \frac{C_R \cap C_D}{C_D} \tag{17}$$

where $C_R$ denotes entities in ground-truth community and $C_D$ denotes entities in community detected by the algorithm.

We calculate *Recall* by dividing the numbers of nodes correctly found by the size of the real-world community. We calculate *Accuracy* by dividing those nodes correctly found by the size of the community detected by the algorithm. *F-Score* considers both of these methods comprehensively.

## Datasets

### Artificial networks

Lancichinetti Fortunato Radicchi (LFR) (*Lancichinetti, Fortunato & Radicchi, 2008*) is a widely used method in complex network research for generating artificial networks that have properties similar to real-world networks. To very the performance of the proposed algorithms and comparison algorithms, three groups of artificial networks generated by *LFR* are used. *LFR* generates different artificial networks by setting these parameters: $\mu$ is a mixing parameter that describes the difficulty of describing the network structure. The greater $\mu$ is, the more difficult it is to describe the community structure. $\overline{|C|}_{min}$ represents the minimum community size in the network; $\overline{d}$ represents the mean degree of node and $d_{max}$ represents the maximum degree of node; $O_n$ represents the number of overlapping nodes and *Om* represents the overlap times of each overlapping node.

This study employs the control variable method to test the performance of the proposed algorithms and the comparison algorithms with different parameters. In this experiment, we change only one parameter at a time. Table 4 lists the settings of artificial networks generated by *LFR*, where the expression [a: b: c] are the value of parameter ranges from a to c with a span of b. The artificial network with a series of parameter $\mu$ is represented by *LFR-$\mu$*; that with a series of parameters $\overline{d}$ and $d_{max}$ is represented by *LFR-$\alpha_{size}$*; and that with a series of parameters $\overline{|C|}_{min}$ and $\overline{|C|}_{max}$ is represented by *LFR-$\alpha_{degree}$*. To ensure experimental precision, we use *LFR* to generate 10 artificial networks under each set of parameters and calculate the average value of each group of results.

**Table 4 The parameter configuration for *LFR* benchmark network.**

| Network | n | $\overline{d}$ | $d_{max}$ | $\overline{|C|}_{min}$ | $\overline{|C|}_{max}$ | μ | $O_n$ | $O_m$ |
|---|---|---|---|---|---|---|---|---|
| LFR-μ | 1,000 | 5 | 25 | 5 | 100 | [0.1:0.1:0.8] | 0 | 0 |
| LFR-$\alpha_{size}$ | 1,000 | 5 | 25 | [10:5:30] | 10 × [10:5:30] | 0.1 | 0 | 0 |
| LFR-$\alpha_{degree}$ | 1,000 | [5:1:10] | 10 × [5:1:10] | 5 | 100 | 0.1 | 0 | 0 |

***Real-world networks***

Table 5 displays the characteristics of six widely used real-world networks involved in this study. The Karate Club network (*Zachary, 1977*) is the membership network of a karate club in an American university. The Football network is the result of 2,000 American College Football League (*Girvan & Newman, 2002*). *RU*, *EN*, *ES* and *FR* are derived from http://snap.stanford.edu/data/. To ensure completion of the experiment within the specified time, we remove the links with Hub Promoted Index (*HPI*) coefficients less than 0.8 in *RU*, *EN*, *ES* and *FR* of the network. The reason for these results is that for networks *RU*, *EN*, *ES* and *FR* with an *HPI* coefficient less than 0.8 reserved, most local community detection algorithms included in this study could not complete the test within the specified time.

## Experimental settings

In this experiment, we refer to the proposed algorithms based on Definition 6, Definition 7 and Definition 8 as *OIRLCDF*, *OIRLCDS* and *OIRLCDT* respectively. Additionally, each algorithm has a version that uses *SSCS* called Algorithm1 and a version that uses *SSSC* called Algorithm 2. For example, *OIRLCDF1* and *OIRLCDF2* represent the algorithms that use *SSCS* and those that use *SSSC*, respectively. During the comparative experiments, we only replace the similarity indices of node similarity in the proposed algorithms. We name the corresponding algorithms based on the similarity index used. Three commonly used similarity indices involved in this experiment are the Jaccard similarity index (*Jaccard, 1901*), Salton similarity index (*Salton & McGill, 1986*) and *RA* similarity index (*Zhou, Lü & Zhang, 2009*). Three novel complex similarity indices involved in this experiment are *CS* (*Zhou, Lü & Zhang, 2009*), *TNS* (*Xu, Guo & Yang, 2020*) and CN (*Liu et al., 2022*).

In addition, we compare *OIRLCD* to five existing local community detection algorithms: LWP (Luo, Wang and Promislow) (*Luo, Wang & Promislow, 2008*), Chen (*Chen, Zaï & Goebel, 2009*), LS (link similarity) (*Chen, Zaï & Goebel, 2009*), LCD (local community detection based on maximum cliques) (*Wu et al., 2012*) and RTLCD (*Zhang, Ding & Yang, 2019*).

*Luo, Wang & Promislow (2008)* proposed an improved quality function *M* based on the Clauset algorithm (*Clauset, 2005*). *M* is calculated by dividing the inner links of the community by the links between communities. Similar to the Clauset algorithm, the *LWP* algorithm expands the community by optimizing the quality function *M*. To manage outliers, *Chen, Zaï & Goebel (2009)* proposed a local community detection algorithm based on quality function *L*. The *Chen* algorithm rechecks the removed nodes to identify whether

**Table 5 The characteristics of real-world networks.**

| Network | n | m | $\bar{d}$ | μ | $|C|$ | $\overline{|C|}_{max}$ | $O_n$ | $O_m$ |
|---|---|---|---|---|---|---|---|---|
| Karate | 34 | 156 | 4.58 | 0.128 | 2 | 17 | 0 | – |
| Football | 115 | 1,226 | 10.66 | 0.357 | 12 | 35 | 0 | – |
| Musae_RU | 896 | 3,698 | 4.12 | 0.47 | 565 | 233 | 547 | 6.23 |
| Musae_EN | 918 | 1,081 | 2.50 | 0.266 | 560 | 43 | 507 | 3.80 |
| Musae_ES | 1,529 | 6,602 | 5.18 | 0.47 | 978 | 285 | 958 | 6.51 |
| Musae_FR | 2,521 | 13,064 | 5.18 | 0.56 | 1,689 | 589 | 1,670 | 7.39 |

they optimize the quality function; this operation can reduce the effects of outliers. *Wu et al. (2012)* proposed a local community detection algorithm based on link similarity (*LS*) that calculates similarity based on the intersection of adjacent nodes of the node and adjacent nodes of the community. The *LS* algorithm expands communities by optimizing node similarity. *Fanrong et al. (2014)* proposed a local community detection algorithm (*LCD*) based on maximum network cliques. The *LCD* algorithm searches maximum cliques as seeds and expands from these nodes by optimizing the distribution of maximum cliques. *Zhang, Ding & Yang (2019)* proposed a robust community detection algorithm *RTLCD* which consists of two stages: seed selection stage and community expansion stage. During seed selection stage, *RTLCD* searches the core node as the alternative seed of the given node, which solves the seed-dependent problem. In the community expansion stage, *RTLCD* expands the community by node relation strength, which maintains seed validity.

Additionally, the proposed and other algorithms are applied to the dataset mentioned above, with the exception that any algorithm running for more than 24 h is stopped.

## Experimental results on real-world networks

Table 6 describes the performance of the proposed algorithms and five existing local community detection algorithms based on NMI, Recall, Precision, F-score and time metrics in six real-world networks. The best and the second-best values are marked in bold. Table 7 lists the percentage gains in terms of *NMI* and F-Measure for algorithms using *SSCS* compared to algorithms using *SSSC*.

From Table 6, we can observe that *OIRLCDT* outperforms *OIRLCDS* in the NMI, Recall, Precision, F-score metrics; and *OIRLCDS* outperforms *OIRLCDT* in these metrics. Improving the precision of the similarity index improves the performance of algorithms on each metric. This phenomenon demonstrates that enhancing the precision of the similarity index can increase the accuracy of detecting local communities. Notably, *OIRLCDT* outperforms all the other comparison algorithms, except *LCD*, on each metric of all six real-world networks. Therefore, *OIRLCDT* can effectively detect local communities and exhibit better performance than the existing algorithms tested in this study. However, *LCD* can achieve good performance, but it lacks scalability in three large real-world networks; these results indicate that *LCD* is not competitive in large real-world networks. It is further observed that *OIRLCDF*, *OIRLCDS* and *OIRLCDT* show gradual improvement in the time

**Table 6 The results of comparison algorithms on six real-world networks.** The best and the second-best values are marked in bold.

| Karate | OIRLCDF1 | OIRLCDF2 | OIRLCDS1 | OIRLCDS2 | OIRLCDT1 | OIRLCDT2 | Jaccard1 | Jaccard2 |
|---|---|---|---|---|---|---|---|---|
| NMI | 0.9138 | 0.9138 | **1** | **1** | **1** | **1** | 0.669 | 0.669 |
| Recall | 0.9706 | 0.9706 | **1** | **1** | **1** | **1** | 0.8529 | 0.8529 |
| Precision | **1** | **1** | **1** | **1** | **1** | **1** | 0.9723 | 0.9723 |
| F-score | 0.9849 | 0.9849 | **1** | **1** | **1** | **1** | 0.9004 | 0.9004 |
| Time(ms) | 21 | 18 | 18 | 16 | 78 | 28 | 14 | 10 |
| Karate | RA1 | RA2 | CS1 | CS2 | TNS1 | TNS2 | CN1 | CN2 |
| NMI | 0.669 | 0.669 | 0.9138 | 0.9138 | **1** | **1** | 0.8372 | 0.8372 |
| Recall | 0.8529 | 0.8529 | 0.9706 | 0.9706 | **1** | **1** | 0.9706 | 0.9706 |
| Precision | 0.9723 | 0.9723 | **1** | **1** | **1** | **1** | 0.9723 | 0.9723 |
| F-score | 0.9004 | 0.9004 | 0.9849 | 0.9849 | **1** | **1** | 0.9706 | 0.9706 |
| Time(ms) | 14 | 12 | 51 | 45 | 46 | 85 | 24 | 12 |
| Karate | lcd | RTLCD | chen | ls | lwp | | | |
| NMI | 0.4093 | **1** | 0.1552 | 0.1688 | 0.516 | | | |
| Recall | 0.6182 | **1** | 0.2071 | 0.2339 | 0.6912 | | | |
| Precision | 0.8449 | **1** | 0.6345 | 0.5588 | 0.8019 | | | |
| F-score | 0.6918 | **1** | 0.2949 | 0.3171 | 0.7179 | | | |
| Time(ms) | 39 | 22062 | 20 | 7 | 14 | | | |
| Football | OIRLCDF1 | OIRLCDF2 | OIRLCDS1 | OIRLCDS2 | OIRLCDT1 | OIRLCDT2 | Jaccard1 | Jaccard2 |
| NMI | 0.7231 | 0.5714 | 0.7231 | 0.5714 | 0.7148 | 0.5714 | **0.7303** | 0.5853 |
| Recall | 0.749 | 0.6122 | 0.749 | 0.6122 | 0.749 | 0.6122 | 0.7421 | 0.5983 |
| Precision | 0.8071 | 0.5998 | 0.8071 | 0.5998 | **0.8002** | 0.5998 | 0.8106 | 0.6068 |
| F-score | 0.7674 | 0.6038 | 0.7674 | 0.6038 | 0.7635 | 0.6038 | **0.7666** | 0.6023 |
| Time(ms) | 179 | 149 | 267 | 239 | 393 | 358 | 146 | 121 |
| Football | RA1 | RA2 | CS1 | CS2 | TNS1 | TNS2 | CN1 | CN2 |
| NMI | **0.7303** | 0.5853 | 0.6412 | 0.4876 | **0.7303** | 0.5853 | 0.717 | 0.5853 |
| Recall | 0.7421 | 0.5983 | 0.6551 | 0.5026 | 0.7421 | 0.5983 | 0.729 | 0.5983 |
| Precision | 0.8106 | 0.6068 | 0.8106 | 0.6068 | 0.8106 | 0.6068 | 0.8106 | 0.6068 |
| F-score | **0.7666** | 0.6023 | 0.7087 | 0.5385 | **0.7666** | 0.6023 | 0.7579 | 0.6023 |
| Time(ms) | 154 | 128 | 393 | 263 | 658 | 567 | 180 | 146 |
| Football | lcd | RTLCD | chen | ls | lwp | | | |
| NMI | 0.5638 | 0.5146 | 0.5863 | 0.5714 | 0.6023 | | | |
| Recall | 0.728 | **0.9209** | 0.6665 | 0.5956 | 0.6409 | | | |
| Precision | 0.6354 | 0.5568 | 0.6456 | 0.6461 | 0.6257 | | | |
| F-score | 0.6708 | 0.6639 | 0.6479 | 0.618 | 0.6301 | | | |
| Time(ms) | 223 | 311 | 297 | 37 | 36 | | | |
| Musae_EN | OIRLCDF1 | OIRLCDF2 | OIRLCDS1 | OIRLCDS2 | OIRLCDT1 | OIRLCDT2 | Jaccard1 | Jaccard2 |
| NMI | 0.5791 | 0.5788 | 0.632 | 0.6284 | **0.6374** | 0.6332 | 0.5983 | 0.593 |
| Recall | 0.6446 | 0.6571 | 0.828 | 0.8396 | **0.8587** | **0.8704** | 0.6655 | 0.6709 |
| Precision | **0.7414** | 0.7273 | 0.6866 | 0.674 | 0.6805 | 0.6671 | 0.7277 | 0.7149 |
| F-score | 0.6447 | 0.6455 | 0.7014 | 0.6989 | **0.7058** | 0.7026 | 0.6636 | 0.6592 |
| Time(ms) | 123 | 103 | 172 | 133 | 200 | 165 | 122 | 103 |

| Karate | OIRLCDF1 | OIRLCDF2 | OIRLCDS1 | OIRLCDS2 | OIRLCDT1 | OIRLCDT2 | Jaccard1 | Jaccard2 |
|---|---|---|---|---|---|---|---|---|
| musae_EN | RA1 | RA2 | CS1 | CS2 | TNS1 | TNS2 | CN1 | CN2 |
| NMI | 0.5935 | 0.592 | 0.489 | 0.4859 | 0.6242 | 0.6233 | 0.5977 | 0.5929 |
| Recall | 0.6534 | 0.6646 | 0.58 | 0.588 | 0.761 | 0.7718 | 0.669 | 0.6758 |
| Precision | 0.7331 | 0.7176 | 0.7341 | 0.7226 | 0.7185 | 0.7044 | 0.7271 | 0.7147 |
| F-score | 0.6586 | 0.6584 | 0.568 | 0.5656 | 0.6915 | 0.6915 | 0.6633 | 0.6595 |
| Time(ms) | 133 | 109 | 182 | 151 | 292 | 194 | 116 | 100 |
| musae_EN | lcd | RTLCD | chen | ls | lwp | | | |
| NMI | **0.6919** | 0.5867 | 0.3476 | 0.5784 | 0.6138 | | | |
| Recall | 0.8544 | 0.8346 | 0.3736 | 0.6592 | 0.7359 | | | |
| Precision | **0.7496** | 0.6663 | 0.468 | 0.6777 | 0.6613 | | | |
| F-score | **0.7652** | 0.6633 | 0.4022 | 0.6338 | 0.6696 | | | |
| Time(ms) | 270 | 0 | 436 | 85 | 82 | | | |
| musae_ES | OIRLCDF1 | OIRLCDF2 | OIRLCDS1 | OIRLCDS2 | OIRLCDT1 | OIRLCDT2 | Jaccard1 | Jaccard2 |
| NMI | 0.3207 | 0.3057 | 0.3437 | 0.3282 | **0.3445** | 0.3288 | 0.3242 | 0.3029 |
| Recall | 0.592 | 0.6004 | 0.751 | 0.7694 | 0.7666 | **0.788** | 0.5354 | 0.5302 |
| Precision | 0.4838 | 0.4412 | 0.4397 | 0.3968 | 0.4337 | 0.3913 | **0.4862** | 0.4446 |
| F-score | 0.4217 | 0.4058 | 0.4456 | 0.4294 | **0.4465** | 0.43 | 0.4248 | 0.4018 |
| Time(ms) | 35,588 | 33,979 | 36,132 | 33,823 | 70,832 | 68,487 | 38,523 | 36,521 |
| musae_ES | RA1 | RA2 | CS1 | CS2 | TNS1 | TNS2 | CN1 | CN2 |
| NMI | 0.3268 | 0.3111 | 0.2992 | 0.2856 | 0.3426 | 0.3293 | 0.334 | 0.3164 |
| Recall | 0.5571 | 0.5626 | 0.6428 | 0.6626 | 0.6732 | 0.6906 | 0.5972 | 0.6018 |
| Precision | 0.4889 | 0.4441 | 0.4584 | 0.4151 | 0.464 | 0.4183 | 0.48 | 0.4379 |
| F-score | 0.4269 | 0.4101 | 0.4029 | 0.3889 | 0.4376 | 0.4238 | 0.4334 | 0.4149 |
| Time(ms) | 34,623 | 32,025 | 39,846 | 37,721 | 108,327 | 111,225 | 37,279 | 35,310 |
| musae_ES | lcd | RTLCD | chen | ls | lwp | | | |
| NMI | **0.4212** | 0.3125 | 0.2094 | 0.2356 | 0.218 | | | |
| Recall | 0.6752 | **0.7916** | 0.2438 | 0.2461 | 0.2952 | | | |
| Precision | 0.5734 | 0.3434 | 0.3397 | 0.4401 | 0.2586 | | | |
| F-score | **0.5374** | 0.3784 | 0.255 | 0.2798 | 0.2518 | | | |
| Time(ms) | 2,888,734 | 0 | 161,900 | 5,244 | 1,246 | | | |
| musae_FR | OIRLCDF1 | OIRLCDF2 | OIRLCDS1 | OIRLCDS2 | OIRLCDT1 | OIRLCDT2 | Jaccard1 | Jaccard2 |
| NMI | 0.198 | 0.1898 | 0.2202 | 0.2125 | **0.2222** | 0.2141 | 0.2127 | 0.2012 |
| Recall | 0.5928 | 0.6093 | 0.6936 | 0.7124 | 0.707 | **0.7212** | 0.4812 | 0.4867 |
| Precision | 0.3715 | 0.34 | 0.3607 | 0.3287 | 0.3552 | 0.3251 | **0.4017** | 0.3664 |
| F-score | 0.3225 | 0.3148 | 0.3477 | 0.3404 | **0.3501** | 0.3423 | 0.3382 | 0.3274 |
| Time(ms) | 908,895 | 876,780 | 698,960 | 661,200 | 1,541,239 | 1,497,477 | 713,875 | 676,211 |
| musae_FR | RA1 | RA2 | CS1 | CS2 | TNS1 | TNS2 | CN1 | CN2 |
| NMI | 0.2074 | 0.2017 | 0.1947 | 0.1873 | **0.2265** | 0.2207 | 0.2176 | 0.208 |
| Recall | 0.445 | 0.4562 | 0.5403 | 0.5545 | 0.5552 | 0.5699 | 0.5125 | 0.5213 |
| Precision | **0.413** | 0.3769 | 0.3869 | 0.3523 | **0.3981** | 0.3635 | 0.4006 | 0.3653 |
| F-score | 0.3235 | 0.319 | 0.3136 | 0.3067 | **0.3452** | 0.3403 | 0.3367 | 0.3277 |

(Continued)

| Karate | OIRLCDF1 | OIRLCDF2 | OIRLCDS1 | OIRLCDS2 | OIRLCDT1 | OIRLCDT2 | Jaccard1 | Jaccard2 |
|---|---|---|---|---|---|---|---|---|
| Time(ms) | 748,261 | 723,873 | 1,021,364 | 996,331 | 4,132,092 | 3,939,218 | 710,195 | 703,725 |
| musae_FR | lcd | RTLCD | chen | ls | lwp | | | |
| NMI | 0 | 0.1875 | 0.1326 | 0.1399 | 0.1189 | | | |
| Recall | 0 | **0.8087** | 0.1469 | 0.1409 | 0.1683 | | | |
| Precision | 0 | 0.2921 | 0.255 | 0.3595 | 0.1387 | | | |
| F-score | 0 | 0.3138 | 0.1627 | 0.1713 | 0.1363 | | | |
| Time(ms) | – | 0 | 1,493,458 | 115,455 | 17,136 | | | |
| musae_RU | OIRLCDF1 | OIRLCDF2 | OIRLCDS1 | OIRLCDS2 | OIRLCDT1 | OIRLCDT2 | Jaccard1 | Jaccard2 |
| NMI | 0.348 | 0.3328 | 0.3929 | 0.3815 | **0.3987** | 0.3812 | 0.3648 | 0.3437 |
| Recall | 0.4981 | 0.5015 | 0.765 | **0.8326** | 0.7996 | **0.8425** | 0.476 | 0.4776 |
| Precision | 0.5625 | 0.4939 | 0.4915 | 0.4372 | 0.4816 | 0.433 | 0.5657 | 0.4991 |
| F-score | 0.4438 | 0.4264 | 0.5023 | 0.4907 | **0.5114** | 0.4894 | 0.4524 | 0.4299 |
| Time(ms) | 13,533 | 13,942 | 12,646 | 13,124 | 26,046 | 26,354 | 13,620 | 13,935 |
| musae_RU | RA1 | RA2 | CS1 | CS2 | TNS1 | TNS2 | CN1 | CN2 |
| NMI | 0.3582 | 0.3421 | 0.3224 | 0.3111 | 0.3769 | 0.3651 | 0.3634 | 0.3446 |
| Recall | 0.4631 | 0.4701 | 0.4856 | 0.4975 | 0.5483 | 0.563 | 0.4924 | 0.4988 |
| Precision | **0.5712** | 0.5014 | 0.5429 | 0.4867 | 0.5585 | 0.4869 | 0.5634 | 0.4939 |
| F-score | 0.4455 | 0.4282 | 0.4255 | 0.4124 | 0.4733 | 0.4605 | 0.4558 | 0.4361 |
| Time(ms) | 9,934 | 9,345 | 15,003 | 15,332 | 51,548 | 56,526 | 13,906 | 14,138 |
| musae_RU | lcd | RTLCD | chen | ls | lwp | | | |
| NMI | **0.4573** | 0.3637 | 0.2045 | 0.3088 | 0.3156 | | | |
| Recall | 0.6982 | 0.8229 | 0.2253 | 0.3478 | 0.4209 | | | |
| Precision | **0.6062** | 0.4486 | 0.323 | 0.4925 | 0.3665 | | | |
| F-score | **0.5785** | 0.4851 | 0.2482 | 0.366 | 0.3643 | | | |
| Time(ms) | 1,897,590 | 80,925 | 55,170 | 2,380 | 1,414 | | | |

**Table 7 The percentage of improvement between comparison algorithms on six real-world networks.**

| karate | OIRLCDF | OIRLCDS | OIRLCDT | Jaccard | RA | CS | TNS | CN |
|---|---|---|---|---|---|---|---|---|
| NMI | 0 | 0 | 0 | 0 | 0 | 0 | 0 | 0 |
| F-score | 0 | 0 | 0 | 0 | 0 | 0 | 0 | 0 |
| football | OIRLCDF | OIRLCDS | OIRLCDT | Jaccard | RA | CS | TNS | CN |
| NMI | 15.17 | 15.17 | 14.34 | 14.5 | 16.36 | 16.36 | 15.97 | 16.43 |
| F-score | 14.5 | 15.36 | 14.5 | 13.17 | 16.43 | 17.02 | 16.43 | 15.56 |
| musae_EN | OIRLCDF | OIRLCDS | OIRLCDT | Jaccard | RA | CS | TNS | CN |
| NMI | 0.05 | 0.57 | 0.66 | 0.89 | 0.25 | 0.64 | 0.14 | 0.81 |
| F-score | −0.12 | 0.36 | 0.46 | 0.67 | 0.03 | 0.42 | 0.00 | 0.58 |
| musae_ES | OIRLCDF | OIRLCDS | OIRLCDT | Jaccard | RA | CS | TNS | CN |
| NMI | 4.91 | 4.72 | 4.77 | 7.03 | 5.05 | 4.76 | 4.04 | 5.56 |

| karate | OIRLCDF | OIRLCDS | OIRLCDT | Jaccard | RA | CS | TNS | CN |
|--------|---------|---------|---------|---------|------|------|------|------|
| F-score | 3.92 | 3.77 | 3.84 | 5.72 | 4.10 | 3.60 | 3.26 | 4.46 |
| musae_FR | OIRLCDF | OIRLCDS | OIRLCDT | Jaccard | RA | CS | TNS | CN |
| NMI | 4.32 | 3.62 | 3.78 | 5.72 | 2.83 | 3.95 | 2.63 | 4.62 |
| F-score | 2.45 | 2.14 | 2.28 | 3.30 | 1.41 | 2.25 | 1.44 | 2.75 |
| F-score | 3.86 | 4.02 | 4.66 | 3.75 | 4.17 | 3.23 | 5.71 | 2.83 |
| musae_RU | OIRLCDF | OIRLCDS | OIRLCDT | Jaccard | RA | CS | TNS | CN |
| NMI | 4.57 | 2.99 | 4.59 | 6.14 | 4.71 | 3.63 | 3.23 | 5.46 |
| F-score | 4.08 | 2.36 | 4.50 | 5.23 | 4.04 | 3.18 | 2.78 | 4.52 |

metric, suggesting that it takes more time to enhance the accuracy of similarity indicators in detecting communities.

Table 7 shows that, when using the same similarity index, the algorithm using *SSCS* outperforms the noe using *SSSC*. This result suggests that *SSCS* is more effective than *SSSC* in finding the core node. Table 6 shows that the algorithm using *SSCS* takes more time than the one using *SSSC*. For the *Karate* network, *SSCS* performs the same as *SSSC* because the *Karate* network is so small for different seed selection strategies to make a significant difference.

## Experimental results on artificial networks

### Experimental results on LFR-μ

We evaluated the community identification ability of the algorithms by analyzing their results on the artificial network of *LFR-μ*. The performance of the proposed algorithms and the comparison algorithms on the *NMI* and F-score metrics are presented in Tables 8 and 9. The first column of the tables represents the parameter $\mu$, ranging from 0.1 to 0.8, while the following columns show the performance of each algorithm under the corresponding $\mu$. As demonstrated in Tables 8 and 9, the performance of all algorithms on *NMI* and *F score* metrics declines from top to bottom. This is because the mixing parameter $\mu$ describing the ratio of the number of neighboring nodes of a node outside the community to the number of all neighboring nodes of the node. The greater the value of $\mu$, the more challenging it is to describe the community structure. As $\mu$ increases, the performance of the algorithms declines.

From Tables 8 and 9, we can observe that the performance of *OIRLCDT* is better than that of *OIRLCDS* on *NMI* and *F-score* metrics. Similarly, the performance of *OIRLCDS* is better than that of *OIRLCDSF*. This indicates that improving the precision of the similarity index can lead to better accuracy of detecting local communities. Moreover, the performance of Jaccard and *RA* is lower than that of the proposed algorithm, *CS*, *TNS* and *CN*. This shows that the higher the precision of the similarity index, the more precise the community detection result. However, the improvement in algorithm precision caused by the improvement in similarity index precision decreases with an increase in the parameter $\mu$. This highlights that as the network becomes more complex, the improvement effect of

**Table 8 NMI of algorithms on LFR-µ.** The best and the second-best values are marked in bold.

| µ | OIRLCDF1 | OIRLCDF2 | OIRLCDS1 | OIRLCDS2 | OIRLCDT1 | OIRLCDT2 | Jaccard1 | Jaccard2 |
|---|---|---|---|---|---|---|---|---|
| 0.1 | 0.253 | 0.2406 | 0.3195 | 0.3011 | **0.3656** | 0.3402 | 0.1805 | 0.1657 |
| 0.2 | 0.1529 | 0.1358 | 0.1866 | 0.1639 | **0.2199** | 0.1868 | 0.1003 | 0.0873 |
| 0.3 | 0.1084 | 0.0869 | 0.131 | 0.1045 | **0.1595** | **0.1261** | 0.0662 | 0.0498 |
| 0.4 | 0.0603 | 0.0482 | 0.0743 | 0.0608 | **0.0896** | **0.0698** | 0.0354 | 0.0262 |
| 0.5 | 0.0359 | 0.0284 | 0.0446 | 0.0351 | **0.0539** | **0.0419** | 0.022 | 0.016 |
| 0.6 | 0.0182 | 0.0132 | 0.0233 | 0.018 | **0.0272** | 0.0208 | 0.0146 | 0.0101 |
| 0.7 | 0.0108 | 0.0081 | 0.0131 | 0.0104 | **0.0143** | 0.0116 | 0.0101 | 0.0074 |
| 0.8 | 0.0074 | 0.0055 | 0.0085 | 0.0069 | 0.0089 | 0.0073 | 0.0073 | 0.0052 |
| µ | RA1 | RA2 | CS1 | CS2 | CN1 | CN2 | TNS1 | TNS2 |
| 0.1 | 0.1862 | 0.1719 | 0.1986 | 0.1857 | 0.2102 | 0.1988 | 0.1993 | 0.1887 |
| 0.2 | 0.0992 | 0.0876 | 0.1124 | 0.1007 | 0.1158 | 0.1033 | 0.1053 | 0.0948 |
| 0.3 | 0.0657 | 0.0513 | 0.0792 | 0.0627 | 0.083 | 0.0665 | 0.0692 | 0.0541 |
| 0.4 | 0.0337 | 0.0256 | 0.0419 | 0.0312 | 0.0415 | 0.0323 | 0.0332 | 0.0261 |
| 0.5 | 0.0217 | 0.016 | 0.0261 | 0.0194 | 0.0252 | 0.0187 | 0.0221 | 0.0167 |
| 0.6 | 0.0145 | 0.0099 | 0.0148 | 0.0102 | 0.0153 | 0.0107 | 0.0141 | 0.0099 |
| 0.7 | 0.0101 | 0.0074 | 0.0104 | 0.0078 | 0.0103 | 0.0078 | 0.0101 | 0.0074 |
| 0.8 | 0.0073 | 0.0052 | 0.0073 | 0.0053 | 0.0074 | 0.0053 | 0.0073 | 0.0052 |
| µ | RTLCD | LWP | Chen | LS | LCD | | | |
| 0.1 | **0.3718** | 0.2865 | 0.1075 | 0.0723 | 0.3504 | | | |
| 0.2 | **0.2025** | 0.1313 | 0.0734 | 0.0496 | 0.1985 | | | |
| 0.3 | 0.1182 | 0.0674 | 0.064 | 0.0425 | 0.1169 | | | |
| 0.4 | 0.0642 | 0.0196 | 0.0433 | 0.0189 | 0.0637 | | | |
| 0.5 | 0.0361 | 0.0075 | 0.0325 | 0.0119 | 0.0406 | | | |
| 0.6 | 0.0173 | 0.0044 | 0.0236 | 0.0085 | **0.0256** | | | |
| 0.7 | 0.0101 | 0.0032 | 0.0179 | 0.0069 | **0.0166** | | | |
| 0.8 | 0.0062 | 0.0019 | **0.0131** | 0.0054 | **0.0118** | | | |

**Table 9 F-score of algorithms on LFR-µ.** The best and the second-best values are marked in bold.

| µ | OIRLCDF1 | OIRLCDF2 | OIRLCDS1 | OIRLCDS2 | OIRLCDT1 | OIRLCDT2 | Jaccard1 | Jaccard2 |
|---|---|---|---|---|---|---|---|---|
| 0.1 | 0.2938 | 0.2753 | 0.3697 | 0.3407 | 0.4285 | 0.3912 | 0.2118 | 0.1923 |
| 0.2 | 0.1849 | 0.1596 | 0.2327 | 0.1987 | 0.2863 | 0.2389 | 0.123 | 0.1038 |
| 0.3 | 0.1327 | 0.1042 | 0.1723 | 0.1363 | **0.2235** | 0.1758 | 0.0827 | 0.0609 |
| 0.4 | 0.0764 | 0.059 | 0.1069 | 0.0859 | 0.1417 | 0.1108 | 0.0457 | 0.0331 |
| 0.5 | 0.0462 | 0.0354 | 0.0703 | 0.0566 | 0.0956 | 0.0761 | 0.0287 | 0.0205 |
| 0.6 | 0.0237 | 0.0169 | 0.0414 | 0.0334 | 0.0611 | 0.0529 | 0.0191 | 0.0131 |
| 0.7 | 0.0148 | 0.0112 | 0.0275 | 0.0231 | 0.0432 | 0.0414 | 0.0135 | 0.0098 |
| 0.8 | 0.01 | 0.0076 | 0.0197 | 0.0179 | 0.0336 | 0.0331 | 0.0096 | 0.0067 |
| µ | RA1 | RA2 | CS1 | CS2 | CN1 | CN2 | TNS1 | TNS2 |
| 0.1 | 0.215 | 0.1968 | 0.2283 | 0.2113 | 0.2416 | 0.2259 | 0.2233 | 0.2095 |

| Table 9 (continued) | | | | | | | |
|---|---|---|---|---|---|---|---|
| μ | OIRLCDF1 | OIRLCDF2 | OIRLCDS1 | OIRLCDS2 | OIRLCDT1 | OIRLCDT2 | Jaccard1 | Jaccard2 |
| 0.2 | 0.1196 | 0.1021 | 0.1346 | 0.1171 | 0.1386 | 0.1201 | 0.1244 | 0.1089 |
| 0.3 | 0.0814 | 0.0624 | 0.0959 | 0.0743 | 0.1007 | 0.079 | 0.0836 | 0.064 |
| 0.4 | 0.0433 | 0.0322 | 0.0531 | 0.0388 | 0.0527 | 0.0401 | 0.0422 | 0.0324 |
| 0.5 | 0.028 | 0.0201 | 0.0334 | 0.0243 | 0.0325 | 0.0236 | 0.0283 | 0.0208 |
| 0.6 | 0.0187 | 0.0128 | 0.0193 | 0.0132 | 0.0197 | 0.0138 | 0.0181 | 0.0126 |
| 0.7 | 0.0133 | 0.0096 | 0.0138 | 0.0104 | 0.0137 | 0.0104 | 0.0132 | 0.0097 |
| 0.8 | 0.0094 | 0.0067 | 0.0096 | 0.007 | 0.0096 | 0.0069 | 0.0093 | 0.0066 |
| μ | RTLCD | LWP | Chen | LS | LCD | | | |
| 0.1 | **0.5126** | 0.3322 | 0.1521 | 0.0889 | **0.4427** | | | |
| 0.2 | **0.3467** | 0.1777 | 0.1164 | 0.064 | **0.3036** | | | |
| 0.3 | **0.2439** | 0.1119 | 0.1111 | 0.0575 | 0.2216 | | | |
| 0.4 | **0.1765** | 0.0469 | 0.0878 | 0.0316 | **0.1594** | | | |
| 0.5 | **0.1337** | 0.0263 | 0.0744 | 0.0229 | **0.1235** | | | |
| 0.6 | **0.1033** | 0.0209 | 0.0626 | 0.0187 | **0.0964** | | | |
| 0.7 | **0.0927** | 0.0217 | 0.0542 | 0.017 | **0.0791** | | | |
| 0.8 | **0.0837** | 0.015 | 0.0446 | 0.0139 | **0.0666** | | | |

similarity index precision decreases, while the time consumption markedly increases, as shown in Table 10.

Tables 8 and 9 indicate that when using the same similarity index, the algorithm using *SSCS* outperforms that using *SSSC*. Therefore, *SSCS* is more effective than *SSSC* when finding the core node. Table 10 shows that the time cost of the algorithm using *SSCS* is also higher than that using *SSSC*.

In all artificial networks with different $\mu$, the performance of OIRLCDT2 on each metric is better than that of the comparison algorithms. This indicates that OIRLCDT detects local communities more effectively than the tested existing algorithms.

### Experimental results on LFR-$\alpha_{degree}$

We evaluate the ability of different community identification algorithms to handle diverse node degrees by applying them to artificial networks generated with the *LFR-$\alpha_{degree}$* model. We list the performance of the proposed algorithms and the comparison algorithms on the *NMI* and *F-score* metrics in Tables 11 and 12. The first column of the tables indicate the mean node degree ranging from 10 to 30, while the second column represents the maximum node degree from 100 to 300. The performances of all algorithms on *NMI* and *F score* metrics improves from top to bottom because a greater mean network node degree represents a more diverse node. The more topological information of the node that we can use, the easier the community detection.

Tables 11 and 12 show that the performance of *OIRLCDT* is better than that of *OIRLCDF* in terms of *NMI* and *F-score* metrics, while the performance of *OIRLCDF* is better than that of *OIRLCDS*. This demonstrates that increasing the precision of the

**Table 10 Times(ms) of algorithms on LFR-μ.** The best and the second-best values are marked in bold.

| μ | OIRLCDF1 | OIRLCDF2 | OIRLCDS1 | OIRLCDS2 | OIRLCDT1 | OIRLCDT2 | Jaccard1 | Jaccard2 |
|---|----------|----------|----------|----------|----------|----------|----------|----------|
| 0.1 | 508 | 639 | 624 | 825 | 1,100 | 1,437 | 370 | 433 |
| 0.2 | 552 | 788 | 682 | 940 | 1,273 | 1,775 | 384 | 482 |
| 0.3 | 513 | 691 | 617 | 917 | 1,350 | 2,006 | 388 | 479 |
| 0.4 | 501 | 685 | 619 | 880 | 1,347 | 1,956 | 394 | 448 |
| 0.5 | 458 | 575 | 568 | 808 | 1,225 | 1,866 | 392 | 434 |
| 0.6 | 443 | 485 | 531 | 637 | 1,275 | 1,906 | 393 | 418 |
| 0.7 | 497 | 559 | 586 | 707 | 1,537 | 2,289 | 442 | 489 |
| 0.8 | 482 | 516 | 557 | 640 | 1,508 | 1,981 | 450 | 477 |
| μ | RA1 | RA2 | CS1 | CS2 | CN1 | CN2 | TNS1 | TNS2 |
| 0.1 | 376 | 446 | 574 | 710 | 433 | 549 | 1,095 | 1,463 |
| 0.2 | 382 | 478 | 570 | 767 | 442 | 604 | 1,223 | 1,843 |
| 0.3 | 387 | 469 | 563 | 745 | 447 | 581 | 1,223 | 1,771 |
| 0.4 | 400 | 459 | 536 | 694 | 443 | 518 | 1,257 | 1,626 |
| 0.5 | 390 | 425 | 512 | 590 | 422 | 471 | 1,259 | 1,588 |
| 0.6 | 399 | 423 | 508 | 559 | 422 | 463 | 1,312 | 1,569 |
| 0.7 | 464 | 488 | 580 | 637 | 485 | 531 | 1,558 | 1,938 |
| 0.8 | 448 | 469 | 574 | 607 | 472 | 500 | 1,550 | 1,811 |
| μ | RTLCD | LWP | Chen | LS | LCD | | | |
| 0.1 | 2,880 | 321 | **260** | **50** | 3,417 | | | |
| 0.2 | 9,453 | 315 | **322** | **58** | 3,711 | | | |
| 0.3 | 14,012 | 327 | **358** | **68** | 3,816 | | | |
| 0.4 | 24,507 | 359 | **396** | **72** | 3,139 | | | |
| 0.5 | 36,790 | 321 | **427** | **65** | 3,334 | | | |
| 0.6 | 25,907 | 327 | **433** | **63** | 3,330 | | | |
| 0.7 | 40,920 | 371 | **442** | **72** | 3,997 | | | |
| 0.8 | 27,062 | 338 | **450** | **68** | 3,394 | | | |

similarity index, the more accuracy of detecting local communities. In addition, *CS*, *TNS* and *CN* outperform *Jaccard* and *RA*, which suggests that the higher the precision of the similarity index, the more precise the community detection result. However, When $\bar{d}$ is greater than *20*, the algorithm with the proposed seed selection method performs worse than the algorithm with the previous seed selection method. This is because, as the mean degree of the network increases, the centrality index becomes more important. Therefore, calculating the similarity index first, which leads to a decrease in algorithm precision. Finally, Table 13 shows that the time consumption has significantly increased.

Tables 11 and 12 show that, when using the same similarity index, the algorithm using *SSCS* outperforms the algorithm using *SSSC*. Therefore, *SSCS* is more effective than *SSSC* in finding the core node. Table 13 indicates that the algorithm using *SSCS* also takes longer than the one using *SSSC*.

**Table 11  NMI of algorithms on LFR-α_degree.**

| $\bar{d}$ | $d_{max}$ | OIRLCDF1 | OIRLCDF2 | OIRLCDS1 | OIRLCDS2 | OIRLCDT1 | OIRLCDT2 | Jaccard1 | Jaccard2 |
|---|---|---|---|---|---|---|---|---|---|
| 5 | 50 | 0.253 | 0.2406 | 0.3195 | 0.3011 | 0.3656 | 0.3402 | 0.1805 | 0.1657 |
| 6 | 60 | 0.3688 | 0.3226 | 0.442 | 0.3836 | 0.5126 | 0.4456 | 0.2476 | 0.2173 |
| 7 | 70 | 0.3471 | 0.3145 | 0.4249 | 0.3741 | 0.5069 | 0.4372 | 0.2207 | 0.2041 |
| 8 | 80 | 0.5143 | 0.4109 | 0.6042 | 0.472 | 0.6708 | 0.5032 | 0.346 | 0.2842 |
| 9 | 90 | 0.5568 | 0.4458 | 0.6568 | 0.5195 | 0.7255 | 0.5634 | 0.359 | 0.2912 |
| 10 | 100 | 0.6797 | 0.5086 | 0.7464 | 0.5369 | 0.8182 | 0.5674 | 0.5105 | 0.3837 |
| $\bar{d}$ | $d_{max}$ | RA1 | RA2 | CS1 | CS2 | CN1 | CN2 | TNS1 | TNS2 |
| 5 | 50 | 0.1862 | 0.1719 | 0.1986 | 0.1857 | 0.2102 | 0.1988 | 0.1993 | 0.1887 |
| 6 | 60 | 0.2534 | 0.2215 | 0.2606 | 0.2292 | 0.2806 | 0.2485 | 0.2664 | 0.236 |
| 7 | 70 | 0.2252 | 0.2103 | 0.2346 | 0.2207 | 0.2655 | 0.2519 | 0.2578 | 0.2412 |
| 8 | 80 | 0.372 | 0.3036 | 0.3805 | 0.3057 | 0.3976 | 0.3282 | 0.3973 | 0.3269 |
| 9 | 90 | 0.3651 | 0.3 | 0.3709 | 0.2962 | 0.3953 | 0.3154 | 0.3851 | 0.314 |
| 10 | 100 | 0.5229 | 0.3843 | 0.5329 | 0.4018 | 0.5515 | 0.4166 | 0.57 | 0.4376 |
| $\bar{d}$ | $d_{max}$ | RTLCD | Clauset | LWP | Chen | LS | LCD | | |
| 5 | 50 | 0.3718 | 0.1927 | 0.2865 | 0.1075 | 0.0723 | 0.3504 | | |
| 6 | 60 | 0.4697 | 0.2623 | 0.4988 | 0.1776 | 0.0408 | 0.5049 | | |
| 7 | 70 | 0.4913 | 0.2439 | 0.4584 | 0.1631 | 0.0329 | 0.4878 | | |
| 8 | 80 | 0.504 | 0.3776 | 0.7167 | 0.334 | 0.0537 | 0.636 | | |
| 9 | 90 | 0.5263 | 0.3518 | 0.6653 | 0.3121 | 0.0492 | 0.6244 | | |
| 10 | 100 | 0.5245 | 0.4468 | 0.8102 | 0.484 | 0.0482 | 0.6931 | | |

**Table 12  F-score of algorithms on LFR-α_degree.**

| $\bar{d}$ | $d_{max}$ | OIRLCDF1 | OIRLCDF2 | OIRLCDS1 | OIRLCDS2 | OIRLCDT1 | OIRLCDT2 | Jaccard1 | Jaccard2 |
|---|---|---|---|---|---|---|---|---|---|
| 5 | 50 | 0.2938 | 0.2753 | 0.3697 | 0.3407 | 0.4285 | 0.3912 | 0.2118 | 0.1923 |
| 6 | 60 | 0.4107 | 0.3523 | 0.4842 | 0.414 | 0.5661 | 0.4851 | 0.2713 | 0.2341 |
| 7 | 70 | 0.3956 | 0.3503 | 0.4737 | 0.4091 | 0.568 | 0.4804 | 0.2442 | 0.221 |
| 8 | 80 | 0.5443 | 0.4229 | 0.6349 | 0.4829 | 0.7111 | 0.519 | 0.3583 | 0.2883 |
| 9 | 90 | 0.587 | 0.462 | 0.6873 | 0.5326 | 0.7574 | 0.5758 | 0.3699 | 0.2952 |
| 10 | 100 | 0.6984 | 0.5127 | 0.7647 | 0.5384 | 0.8432 | 0.5712 | 0.5186 | 0.3849 |
| $\bar{d}$ | $d_{max}$ | RA1 | RA2 | CS1 | CS2 | CN1 | CN2 | TNS1 | TNS2 |
| 5 | 50 | 0.215 | 0.1968 | 0.2283 | 0.2113 | 0.2416 | 0.2259 | 0.2233 | 0.2095 |
| 6 | 60 | 0.2756 | 0.2371 | 0.2846 | 0.2461 | 0.3067 | 0.2667 | 0.2843 | 0.2477 |
| 7 | 70 | 0.2492 | 0.2284 | 0.2593 | 0.2391 | 0.2931 | 0.2724 | 0.2771 | 0.255 |
| 8 | 80 | 0.3846 | 0.3094 | 0.3907 | 0.3083 | 0.4109 | 0.3335 | 0.4058 | 0.3292 |
| 9 | 90 | 0.3755 | 0.3036 | 0.3814 | 0.3001 | 0.4076 | 0.3192 | 0.3932 | 0.3171 |
| 10 | 100 | 0.5311 | 0.3858 | 0.5404 | 0.4018 | 0.5589 | 0.4172 | 0.5755 | 0.4358 |
| $\bar{d}$ | $d_{max}$ | RTLCD | Clauset | LWP | Chen | LS | LCD | | |
| 5 | 50 | 0.5126 | 0.2657 | 0.3322 | 0.1521 | 0.0889 | 0.4427 | | |
| 6 | 60 | 0.5973 | 0.3185 | 0.5068 | 0.221 | 0.0455 | 0.5637 | | |

(Continued)

| $\bar{d}$ | $d_{max}$ | OIRLCDF1 | OIRLCDF2 | OIRLCDS1 | OIRLCDS2 | OIRLCDT1 | OIRLCDT2 | Jaccard1 | Jaccard2 |
|---|---|---|---|---|---|---|---|---|---|
| 7 | 70 | 0.6153 | 0.3009 | 0.4668 | 0.2051 | 0.0368 | 0.5478 | | |
| 8 | 80 | 0.6157 | 0.4195 | 0.7218 | 0.3751 | 0.0564 | 0.673 | | |
| 9 | 90 | 0.6423 | 0.3934 | 0.6715 | 0.3503 | 0.052 | 0.662 | | |
| 10 | 100 | 0.6423 | 0.4801 | 0.8152 | 0.5166 | 0.0494 | 0.7188 | | |

**Table 13 Times(ms) of algorithms on LFR-$\alpha_{degree}$.**

| $\bar{d}$ | $d_{max}$ | OIRLCDF1 | OIRLCDF2 | OIRLCDS1 | OIRLCDS2 | OIRLCDT1 | OIRLCDT2 | Jaccard1 | Jaccard2 |
|---|---|---|---|---|---|---|---|---|---|
| 5 | 50 | 498 | 641 | 617 | 822 | 1,060 | 1,406 | 358 | 417 |
| 6 | 60 | 1,278 | 1,708 | 1,529 | 2,067 | 2,980 | 3,839 | 751 | 956 |
| 7 | 70 | 1,419 | 1,826 | 1,658 | 2,068 | 3,480 | 4,483 | 819 | 1,096 |
| 8 | 80 | 2,867 | 3,661 | 3,455 | 4,266 | 7,017 | 8,796 | 1,717 | 2,303 |
| 9 | 90 | 3,791 | 5,256 | 4,391 | 5,513 | 8,935 | 11,271 | 2,060 | 2,981 |
| 10 | 100 | 6,189 | 7,937 | 6,891 | 8,216 | 15,287 | 17,512 | 4,033 | 4,853 |
| $\bar{d}$ | $d_{max}$ | RA1 | RA2 | CS1 | CS2 | CN1 | CN2 | TNS1 | TNS2 |
| 5 | 50 | 361 | 440 | 553 | 699 | 419 | 539 | 1,075 | 1,459 |
| 6 | 60 | 764 | 979 | 1,202 | 1,636 | 890 | 1,155 | 2,974 | 4,436 |
| 7 | 70 | 859 | 1,168 | 1,344 | 1,875 | 1,019 | 1,414 | 3,590 | 5,346 |
| 8 | 80 | 1,784 | 2,410 | 3,034 | 4,211 | 2,139 | 2,836 | 8,840 | 12,865 |
| 9 | 90 | 2,092 | 2,879 | 3,476 | 4,823 | 2,471 | 3,405 | 10,460 | 15,391 |
| 10 | 100 | 4,022 | 4,782 | 7,476 | 10,028 | 4,750 | 6,196 | 24,352 | 34,385 |
| $\bar{d}$ | $d_{max}$ | RTLCD | Clauset | LWP | Chen | LS | LCD | | |
| 5 | 50 | 2,935 | 386 | 312 | 266 | 51 | 3,428 | | |
| 6 | 60 | 10,234 | 1,366 | 931 | 1,185 | 62 | 26,707 | | |
| 7 | 70 | 13,970 | 1,523 | 946 | 1,168 | 68 | 29,684 | | |
| 8 | 80 | 34,977 | 5,896 | 1,689 | 8,247 | 78 | 90,802 | | |
| 9 | 90 | 28,342 | 5,661 | 1,715 | 7,472 | 86 | 115,424 | | |
| 10 | 100 | 52,847 | 14,396 | 2,344 | 31,798 | 106 | 217,044 | | |

In all artificial networks with different value of parameter $\bar{d}$, OIRLCDT2 outperforms the other comparison algorithms across all metrics. This result shows that the proposed algorithm can effectively perform local community detection and is superior to existing algorithms tested in this study.

### Experimental results on LFR-$\alpha_{size}$

We evaluated the ability of community identification algorithms to handle diverse community structures by testing them on the artificial networks of LFR-$\alpha_{size}$. The performance of the proposed algorithms and the comparison algorithms was measured using the *NMI* and *F-score* metrics, and the results are presented in Tables 14 and 15. The first column of the tables shows the minimum community size (ranging from 10 to 30),

**Table 14 NMI of algorithms on LFR-α<sub>size</sub>.**

| $|C|_{min}$ | $|C|_{max}$ | OIRLCDF1 | OIRLCDF2 | OIRLCDS1 | OIRLCDS2 | OIRLCDT1 | OIRLCDT2 | Jaccard1 | Jaccard2 |
|---|---|---|---|---|---|---|---|---|---|
| 10 | 100 | 0.253 | 0.2406 | 0.3195 | 0.3011 | 0.3656 | 0.3402 | 0.1805 | 0.1657 |
| 15 | 150 | 0.1404 | 0.1324 | 0.1872 | 0.184 | 0.231 | 0.225 | 0.0945 | 0.0865 |
| 20 | 200 | 0.0799 | 0.0802 | 0.1227 | 0.1226 | 0.1592 | 0.1637 | 0.0532 | 0.0526 |
| 25 | 250 | 0.0695 | 0.08 | 0.1005 | 0.1166 | 0.1334 | 0.1481 | 0.0364 | 0.0381 |
| 30 | 300 | 0.0434 | 0.0493 | 0.07 | 0.0796 | 0.1052 | 0.1181 | 0.0217 | 0.0204 |
| $|C|_{min}$ | $|C|_{max}$ | RA1 | RA2 | CS1 | CS2 | CN1 | CN2 | TNS1 | TNS2 |
| 10 | 100 | 0.1862 | 0.1719 | 0.1986 | 0.1857 | 0.2102 | 0.1988 | 0.1993 | 0.1887 |
| 15 | 150 | 0.0947 | 0.0887 | 0.1028 | 0.0935 | 0.1091 | 0.1008 | 0.1059 | 0.0982 |
| 20 | 200 | 0.051 | 0.0507 | 0.0542 | 0.0545 | 0.0577 | 0.0583 | 0.0526 | 0.053 |
| 25 | 250 | 0.0371 | 0.0401 | 0.0443 | 0.0487 | 0.0478 | 0.0522 | 0.0412 | 0.0454 |
| 30 | 300 | 0.0215 | 0.021 | 0.025 | 0.0247 | 0.0269 | 0.027 | 0.0242 | 0.0248 |
| $|C|_{min}$ | $|C|_{max}$ | RTLCD | Clauset | LWP | Chen | LS | LCD | | |
| 10 | 100 | 0.3718 | 0.1927 | 0.2865 | 0.1075 | 0.0723 | 0.3504 | | |
| 15 | 150 | 0.2991 | 0.1253 | 0.1836 | 0.0711 | 0.0338 | 0.2331 | | |
| 20 | 200 | 0.2234 | 0.0869 | 0.1232 | 0.0486 | 0.0208 | 0.1679 | | |
| 25 | 250 | 0.205 | 0.0696 | 0.0971 | 0.0397 | 0.015 | 0.1375 | | |
| 30 | 300 | 0.1733 | 0.0468 | 0.0532 | 0.028 | 0.0079 | 0.0935 | | |

and the second column is the maximum size of the community (ranging from 100 to 300). As the community size increases, the structure of the network becomes more diverse, making community detection more challenging. Thus, the performance of all algorithms on *NMI* and *F score* metrics worsens from top to bottom. The reason for this phenomenon is as follows. As the maximum and minimum size of communities in a networkd increase, the community structure becomes more diverse. This increased diversity in the community structure makes community detection more difficult.

Tables 14 and 15 demonstrate that the performance of *OIRLCDT* is better than that of *OIRLCDS* in terms of *NMI* and F-score metrics, and the performance of *OIRLCDS* is better than that of *OIRLCDF*. Thus, increasing the precision of the similarity index improves the performance of algorithms on each metric. This phenomenon highlights that improving the precision of similarity helps to increase the precision of the local community detection algorithm. Furthermore, the performance of Jaccard and *RA* is lower than that of the proposed algorithm, *CS*, *TNS* and *CN*. This shows that the higher the precision of similarity index, the more precise the community detection result. As the community size increases, the algorithm using *SSSC* outperforms the one using *SSCS*. This phenomenon occurs because the similarity index become more important as the community size increase. Therefore, calculating the similarity index first is more effective than calculating the centrality index first, leading to an increase in algorithm precision. However, this also markedly increases the time consumption, as shown in Table 16.

Tables 14 and 15 show that when using the same similarity index, the algorithm using *SSCS* performs better than that using *SSSC*. This result indicates that *SSCS* is more effective

**Table 15  F-score of algorithms on LFR-α_{size}.**

| $|C|_{min}$ | $|C|_{max}$ | OIRLCDF1 | OIRLCDF2 | OIRLCDS1 | OIRLCDS2 | OIRLCDT1 | OIRLCDT2 | Jaccard1 | Jaccard2 |
|---|---|---|---|---|---|---|---|---|---|
| 10 | 100 | 0.2938 | 0.2753 | 0.3697 | 0.3407 | 0.4285 | 0.3912 | 0.2118 | 0.1923 |
| 15 | 150 | 0.1685 | 0.1575 | 0.2265 | 0.2214 | 0.2897 | 0.2791 | 0.1156 | 0.1035 |
| 20 | 200 | 0.1023 | 0.1004 | 0.1583 | 0.1561 | 0.2121 | 0.2137 | 0.0678 | 0.0655 |
| 25 | 250 | 0.0886 | 0.0992 | 0.1326 | 0.1498 | 0.1858 | 0.202 | 0.049 | 0.05 |
| 30 | 300 | 0.0569 | 0.0602 | 0.0984 | 0.1071 | 0.1522 | 0.1642 | 0.0307 | 0.0272 |
| $|C|_{min}$ | $|C|_{max}$ | RA1 | RA2 | CS1 | CS2 | CN1 | CN2 | TNS1 | TNS2 |
| 10 | 100 | 0.215 | 0.1968 | 0.2283 | 0.2113 | 0.2416 | 0.2259 | 0.2233 | 0.2095 |
| 15 | 150 | 0.1146 | 0.1052 | 0.1234 | 0.11 | 0.1311 | 0.1188 | 0.1235 | 0.1122 |
| 20 | 200 | 0.0642 | 0.0627 | 0.0684 | 0.0677 | 0.0727 | 0.0721 | 0.0653 | 0.0644 |
| 25 | 250 | 0.0489 | 0.0517 | 0.0571 | 0.061 | 0.0608 | 0.0645 | 0.051 | 0.0544 |
| 30 | 300 | 0.0299 | 0.0272 | 0.0335 | 0.0311 | 0.036 | 0.0337 | 0.0316 | 0.0299 |
| $|C|_{min}$ | $|C|_{max}$ | RTLCD | Clauset | LWP | Chen | LS | LCD | | |
| 10 | 100 | 0.5126 | 0.2657 | 0.3322 | 0.1521 | 0.0889 | 0.4427 | | |
| 15 | 150 | 0.4538 | 0.1882 | 0.2305 | 0.1066 | 0.0454 | 0.326 | | |
| 20 | 200 | 0.3982 | 0.1408 | 0.167 | 0.0781 | 0.0297 | 0.2535 | | |
| 25 | 250 | 0.3744 | 0.1191 | 0.137 | 0.0667 | 0.0223 | 0.218 | | |
| 30 | 300 | 0.3539 | 0.0873 | 0.083 | 0.0506 | 0.0126 | 0.1658 | | |

**Table 16  Times(ms) of algorithms on LFR-α_{size}.**

| $|C|_{min}$ | $|C|_{max}$ | OIRLCDF1 | OIRLCDF2 | OIRLCDS1 | OIRLCDS2 | OIRLCDT1 | OIRLCDT2 | Jaccard1 | Jaccard2 |
|---|---|---|---|---|---|---|---|---|---|
| 10 | 100 | 508 | 639 | 624 | 825 | 1,100 | 1,437 | 370 | 433 |
| 15 | 150 | 434 | 565 | 546 | 789 | 1,002 | 1,424 | 319 | 366 |
| 20 | 200 | 457 | 578 | 579 | 777 | 1,137 | 1,557 | 351 | 433 |
| 25 | 250 | 491 | 625 | 592 | 825 | 1,151 | 1,625 | 357 | 424 |
| 30 | 300 | 485 | 733 | 659 | 945 | 1,342 | 2,000 | 380 | 482 |
| $|C|_{min}$ | $|C|_{max}$ | RA1 | RA2 | CS1 | CS2 | CN1 | CN2 | TNS1 | TNS2 |
| 10 | 100 | 376 | 446 | 574 | 710 | 433 | 549 | 1,095 | 1,463 |
| 15 | 150 | 327 | 401 | 482 | 600 | 362 | 444 | 966 | 1,325 |
| 20 | 200 | 344 | 413 | 506 | 620 | 383 | 489 | 1,034 | 1,367 |
| 25 | 250 | 363 | 434 | 518 | 660 | 398 | 498 | 1,092 | 1,503 |
| 30 | 300 | 381 | 489 | 522 | 714 | 417 | 546 | 1,199 | 1,797 |
| $|C|_{min}$ | $|C|_{max}$ | RTLCD | Clauset | LWP | Chen | LS | LCD | | |
| 10 | 100 | 2,880 | 375 | 321 | 260 | 50 | 3,417 | | |
| 15 | 150 | 3,750 | 390 | 413 | 269 | 51 | 5,004 | | |
| 20 | 200 | 4,987 | 421 | 396 | 301 | 55 | 4,957 | | |
| 25 | 250 | 5,931 | 440 | 386 | 318 | 55 | 4,581 | | |
| 30 | 300 | 10,301 | 415 | 390 | 347 | 53 | 5,457 | | |

than *SSSC* in finding the core node. Table 16 shows that the time cost of the algorithm using *SSCS* also takes more time than that using *SSSC*.

In all artificial networks with different $\alpha_{size}$, *OIRLCDT2* outperforms the other comparison algorithms on each metric. These results show that the proposed algorithm can effectively perform local community detection better than existing algorithms tested in this study.

## CONCLUSION

This study proposes a novel local community detection algorithm called *OIRLCD*, based on the optimization of the interaction relationships between nodes rather than using the quality function. First, during seed selection process, a novel seed selection method is used to search for the alternative seeds of the given node. This method iteratively searches the most similar neighbor node of the given node, which has the greater node centrality than the given node. The final result is taken as the seed. Second, in the community expansion process, a novel similarity index to used measure the interaction relationship between nodes and community, and communities are expanded communities by adding the node with the most significant interaction relationship to the community.

The proposed similarity index to added to the same algorithm with the other three basic similarity indices and the three latest similarity indices. The proposed algorithm is then compared with five existing local community algorithms in both real-world networks and artificial networks. Experimental results show that the optimization of interaction relationship algorithms based on node similarity can detect communities accurately and efficiently, and a good similarity index can highlight the advantages of the algorithm based on interaction optimization. In addition, the advantages of algorithms with the precision similarity index decrease with the increasing network complexity and are not affected by the parameter mean degree and community size of the network. The advantages of algorithms with the proposed *SSCS* decreases as the parameter mean degree increases; increases as the parameter community size increases; and are not affected by network complexity.

However, there are still some areas that need optimization in this field of study, including finding the optimal balance of time consumption and similarity precision.

### Funding
The authors received no funding for this work.

### Competing Interests
The authors declare that they have no competing interests.

### Author Contributions
- Shenglong Wang conceived and designed the experiments, performed the experiments, analyzed the data, performed the computation work, prepared figures and/or tables, authored or reviewed drafts of the article, and approved the final draft.

- Jing Yang conceived and designed the experiments, performed the experiments, analyzed the data, performed the computation work, prepared figures and/or tables, authored or reviewed drafts of the article, and approved the final draft.
- Xiaoyu Ding conceived and designed the experiments, performed the experiments, analyzed the data, performed the computation work, prepared figures and/or tables, authored or reviewed drafts of the article, and approved the final draft.
- Meng Zhao conceived and designed the experiments, performed the experiments, analyzed the data, performed the computation work, prepared figures and/or tables, authored or reviewed drafts of the article, and approved the final draft.

### Data Availability
The raw data are available in the Supplemental File. The third-party data is available at Twitch Social Networks: https://snap.stanford.edu/data/twitch-social-networks.html.

### Supplemental Information
Supplemental information for this article can be found online at http://dx.doi.org/10.7717/peerj-cs.1386#supplemental-information.

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
