# Peer review of "Detecting local communities in complex network via the optimization of interaction relationship between node and community"

_PeerJ Computer Science, doi:10.7717/peerj-cs.1386_

## Round 0.1 · original submission · Major Revisions

Based on reviewers comments the manuscript needs revisions.

Reviewer 1 ·

Basic reporting

In this research, there is no order for the contents in different sections. It is difficult to read the text due to the fragmentation. It is better to make an orderly arrangement of the background of work and challenges. The model system should be better shown with a block diagram or representational figure. Limitations and conditions should be mentioned in the proposed innovation or solution, etc...

Experimental design

method and proposed scheme should be explained better in formula and illustration....

Validity of the findings

novelty is not clear and is not figured out in a good way..it is hard for reader to understand the step by step method...

Reviewer 2 ·

Basic reporting

The paper proposes a new algorithm to detect communities locally in social networks by introducing new similarity functions. The algorithm is compared with other state of the art algorithms and shows improvements. The paper is well structured and the related works are well-explained. However, the English can be improved. Also, there are some questions and comments which need to be addressed.

1. I suppose the two definitions 1 and 2 define the same concept? If not, please add more details.
2. It would be more thoughtful if you could add written explanations to the definitions.
3. Please use | instead of / in formulas.
4. Please review the indexing of the definitions.
5. Line 246 - Definition 9 - I could not find NS(u,v) defined in the paper?
6. Line 3 - Calculate the degree of each node based on Definition 4Definition 5, - 4 or 5?
7. Line 9 is redundant.
8. All the tables fell to the end of the paper. The tables must appear where they are being referenced.
9. Please reduce the size of the texts on the tables, so the cells fit perfectly – for example you have the number 1000 in two lines in table 4.
10. Please indicate the best scores in tables by making them bold or other ways.
11. Please use a united structure for all tables.
12. Please specify the unit of time reported for algorithms.

Experimental design

The research problem is well-defined. However, there are some questions on the algorithms proposed and the experiments:

1. About the termination condition of the Seed selection process in Algorithm 1, what if the v_seed is the local central node and is the core node of the community itself. In this case the while loop never stops.
2. How did you indicate the ground-truth communities of the real-word networks to calculate F-score.

Validity of the findings

The proposed algorithm shows improvements. The findings seems to be valid. Also, there is a comment on the improvement of the paper:

1. The experiment has been done on several networks but all the employed networks have the same topology. The main challenge of the community detection problem is to indicate communities in diverse types of social networks. In this way, you can show if the proposed metric can capture the best communities in different networks. So, other types of real-world networks need to be involved in the experiments.

---

## Round 0.2 · Minor Revisions

The topic is very important and interesting, but there are still some concerns that should be addressed.


1. More background description on the mentioned concept to be added.

2. Motivation of this study is not clear, and should be clarified.

3.The language usage throughout this paper need to be improved, the author should do some proofreading on it.
4. I still found \N(v)\ instead of |N(v)|.

Reviewer 2 ·

Basic reporting

The paper is very much improved and the current version seems to be an acceptable article. However, the writing can still be improved. Also, I still found \N(v)\ instead of |N(v)|.

Experimental design

All the questions are addressed.

Validity of the findings

The issue is addressed.

---

## Round 0.3 · accepted · Accept

The manuscript can be accept in the current form.